# 2,6-Diaminopurine as a highly potent corrector of UGA nonsense mutations

Carole Trzaska[1,13], Séverine Amand[2,13], Christine Bailly[2,13], Catherine Leroy[1], Virginie Marchand [3], Evelyne Duvernois-Berthet [4], Jean-Michel Saliou [5], Hana Benhabiles[1], Elisabeth Werkmeister[6], Thierry Chassat[7], Romain Guilbert[7], David Hannebique[7], Anthony Mouray[7], Marie-Christine Copin[1], Pierre-Arthur Moreau [8], Eric Adriaenssens[9], Andreas Kulozik [10], Eric Westhof[11], David Tulasne[1], Yuri Motorin [12], Sylvie Rebuffat[2] & Fabrice Lejeune [1✉]

Nonsense mutations cause about 10% of genetic disease cases, and no treatments are available. Nonsense mutations can be corrected by molecules with nonsense mutation readthrough activity. An extract of the mushroom *Lepista inversa* has recently shown high-efficiency correction of UGA and UAA nonsense mutations. One active constituent of this extract is 2,6-diaminopurine (DAP). In Calu-6 cancer cells, in which *TP53* gene has a UGA nonsense mutation, DAP treatment increases p53 level. It also decreases the growth of tumors arising from Calu-6 cells injected into immunodeficient nude mice. DAP acts by interfering with the activity of a tRNA-specific 2′-O-methyltransferase (FTSJ1) responsible for cytosine 34 modification in tRNA$^{Trp}$. Low-toxicity and high-efficiency UGA nonsense mutation correction make DAP a good candidate for the development of treatments for genetic diseases caused by nonsense mutations.

[1] Univ. Lille, CNRS, Inserm, CHU Lille, Institut Pasteur de Lille, UMR9020–UMR-S 1277, CANTHER – Cancer Heterogeneity, Plasticity and Resistance to Therapies, 59000 Lille, France. [2] Muséum National d'Histoire Naturelle, Centre National de la Recherche Scientifique, Laboratory Molecules of Communication and Adaptation of Microorganisms (MCAM), UMR 7245 CNRS-MNHN, 75005 Paris, France. [3] Next-Generation Sequencing Core Facility, UMS2008 IBSLor CNRS-Université de Lorraine-INSERM, BioPôle, 54505 Vandoeuvre-les-Nancy, France. [4] Muséum National d'Histoire Naturelle, Centre National de la Recherche Scientifique, Laboratoire Physiologie Moléculaire et Adaptation (PhyMA), UMR7221 CNRS-MNHN, 75005 Paris, France. [5] CNRS, INSERM, CHU Lille, Institut Pasteur de Lille, U1019, UMR 8204, CIIL-Centre d'Infection et d'Immunité de Lille, University of Lille, 59000 Lille, France. [6] Cellular Microbiology and Physics of Infection Group, Center for Infection and Immunity of Lille, CNRS UMR8204, INSERM U1019, Institut Pasteur de Lille, Lille Regional Univ. Hosp. Centr., Lille Univ., Lille 59000, France. [7] Institut Pasteur de Lille – PLEHTA (Plateforme d'expérimentation et de Haute Technologie Animale), 59019 Lille, France. [8] Univ. Lille, Fac. Pharmacie Lille, ULR 4515, LGCgE, Laboratoire de Génie Civil et géo-Environnement, 59000 Lille, France. [9] Univ. Lille, CNRS, INSERM, CHU.Lille, Centre Oscar Lambert, UMR9020–UMR1277, CANTHER – Cancer Heterogeneity, Plasticity and Résistance to Therapies, 59000 Lille, France. [10] Department of Pediatric Oncology, Hematology and Immunology, Children's Hospital and Hopp Children's Tumor Center Heidelberg, EMBL/Medical Faculty Molecular Medicine Partnership Unit, 69120 Heidelberg, Germany. [11] Architecture and Reactivity of RNA, Institute of Molecular and Cellular Biology of the CNRS UPR9002/University of Strasbourg, Strasbourg 67084, France. [12] Ingénierie Moléculaire et Physiopathologie Articulaire, UMR7365, CNRS - Université de Lorraine, 54505 Vandoeuvre-les-Nancy, France. [13]These authors contributed equally: Carole Trzaska, Séverine Amand, Christine Bailly. ✉email: fabrice.lejeune@inserm.fr

Nonsense mutations cause about 10% of genetic disease cases[1]. The presence of a premature termination codon (PTC) on an mRNA often activates the mRNA surveillance mechanism called nonsense-mediated mRNA decay (NMD), involving the main NMD factors UPF1, UPF2, and UPF3X (also called UPF3b)[2–6]. NMD can degrade PTC-containing mRNAs, thus preventing synthesis of potentially harmful or nonfunctional truncated proteins. Yet, a fraction of the PTC-containing mRNAs eludes NMD, making it theoretically possible to act on them to correct nonsense mutations[7]. As to date no treatments are available for patients with a genetic disease caused by a nonsense mutation, identifying efficient nonsense mutation correctors is a major public health issue.

At the translational level, PTC recognition is closely related to the mechanism of termination. In eukaryotes, when the ribosome reaches a stop codon, competition occurs at the A site of the ribosome between near-cognate tRNAs and the release factors eRF1 and eRF3. In more than 99.9% of cases, the result of this competition is in favor of release factor recruitment[8]. To ensure codon translation by cognate tRNAs, these undergo various posttranscriptional modifications (including methylation, pseudouridylation, deamination, and diverse complex modifications, especially at positions 34 and 37 of the anticodon) having consequences for genome decoding[9–11]. tRNA anticodon modifications, including 2′-O-methylations, play a major role in efficient, error-free decoding of mRNA codons. Many enzymes involved in tRNA modifications have been clearly linked to human pathologies and diseases[12,13].

PTC readthrough during translation involves incorporation, by the ribosome, of an amino acid at the PTC position, so as to complete synthesis of the peptide chain encoded by the open-reading frame[14–17]. A full-length protein synthesized by PTC readthrough results in a difference of no more than one amino acid between the protein synthesized and the wild-type protein. The protein is functional if the amino acid incorporated at the PTC position is compatible with the function of the protein. Recently, PTC readthrough has been shown to occur in specific cytoplasmic bodies called readthrough bodies, suggesting that a specific environment may favor PTC readthrough[18].

Strategies aiming to correct nonsense mutations and rescue expression of PTC-containing genes involve inhibition of NMD, associated or not with activation of PTC readthrough[14,15,18–20]. Several molecules have been shown to activate readthrough, including aminoglycosides, such as gentamicin and G418 and non-aminoglycosides, such as PTC124/ataluren/Translarna and RTC13/14 (refs. [21–29]). Some of the compounds used, such as G418 and amlexanox, can both inhibit NMD and activate readthrough[19,30]. Unfortunately, most of these molecules are either toxic or promote only low-efficiency correction of nonsense mutations in human cells. To date, only PTC124/ataluren/ Translarna has reached clinical phase II/III, for cystic fibrosis and Duchenne muscular dystrophy. As the results, albeit encouraging, were too limited to offer patients a significant benefit, this molecule has not been approved in the United States for the treatment of nonsense-mutation-related Duchenne muscular dystrophy. In Europe, however, it has received a marketing authorization for this pathology[31,32].

In a recent report, we described a specific screening system for identifying compounds that efficiently correct nonsense mutations in human cells. It uses a firefly reporter gene carrying a nonsense mutation and an intron located more than 55 nucleotides downstream, making the corresponding mRNA subject to NMD[33]. This system, unlike other, previously described screening systems using PTC-containing reporter mRNAs immune to NMD, reproduces the fate of PTC-containing mRNAs in patient cells[21,33–35]. We have used it to show that an extract of the common mushroom *Lepista inversa* (H7) acts as a very efficient corrector of UGA and UAA nonsense mutations in human cells[33].

In the present study, we identify the active compounds responsible for the readthrough activity of H7 extract, one of which is 2,6-diaminopurine (DAP). DAP has been used previously for antileukemia treatment and is reported to exert antiviral[36–39] and miRNA inhibition activity[40]. It is demonstrated here to be an efficient and exclusive corrector of UGA nonsense mutations in both immortalized human cells and an in vivo mouse model. At the molecular level, it is further shown that DAP inhibits $Cm_{34}$ methylation of $tRNA^{Trp}$ by tRNA:methyltransferase 1 (FTSJ1). This protein is the human homolog of yeast TRM7, which also catalyzes formation of $Cm_{32}$ and $Nm_{34}$ in some tRNAs, such as $tRNA^{Phe}$ and $tRNA^{Trp}$ (refs. [41,42]).

## Results

**DAP contributes to the PTC-correcting activity of H7 extract.** An extract (H7) prepared from the mushroom *Lepista inversa* (also currently called *Paralepista flaccida*) has recently been shown to exhibit high UAA and UGA readthrough-promoting activity[33]. To identify the compounds responsible for this activity, a semi-preparative HPLC fractionation protocol was used. Nine fractions (F7–F15) were obtained (Fig. 1a) and tested for PTC-readthrough activity in a system using a PTC-carrying firefly luciferase mRNA subject to NMD[33] (Fig. 1b). DMSO and G418 were used, respectively, as a negative and a positive control. As expected[33], G418 corrected the UGA more efficiently than the UAG nonsense mutation, and only weakly corrected the UAA nonsense mutation. Firefly luciferase activity measurements showed that fractions F13 and F15 corrected the UAA nonsense mutation, while fractions F10, F13, and F15 corrected the UGA nonsense mutation (Fig. 1c). The combined use of nuclear magnetic resonance (NMR) and mass spectrometry allowed identifying the major component in fractions F13 and F15 as clitocine ((2R,3R,4S,5R)-2-[(6-amino-5-nitropyrimidin-4-yl)amino]-5-(hydroxymethyl)oxolane-3,4-diol) (76% and 47%, respectively), known as a potent readthrough molecule[24]. In fraction F10 another molecule, showing exclusive readthrough of the UGA stop codon and present at about 77%, was assigned as DAP on the basis of the $^1H$ and $^{13}C$ NMR and mass fragmentation data (Supplementary Table 1, Supplementary Figs. 1–3) (Fig. 1c and Supplementary Table 2). To make sure the UGA readthrough activity exhibited by fraction F10 was due to DAP and not to some other, minor compound, commercial synthetic DAP (98% purity) was tested for PTC-readthrough activity. It showed exclusive correction of the UGA nonsense mutation confirming that the readthrough activity of fraction F10 was due to the presence of DAP (Fig. 1c and Supplementary Table 2). Given the toxicity issues associated with clitocine[43], we decided to focus on DAP. In all subsequent experiments, we used commercial DAP only.

The UGA readthrough efficiency of DAP, as compared to that of G418 (used as a reference molecule), was then tested on the firefly luciferase construct (Fig. 1d and Supplementary Table 3). When added in increasing amount to the cell culture medium G418 caused a slight increase in luciferase activity (to about twice the background level when added at 300 µM). DAP displayed a much higher UGA readthrough efficiency.

**DAP increases readthrough on endogenous p53 UGA mutation.** To further assess its nonsense-mutation-correcting efficiency, DAP was added to the culture medium of cancer cell lines harboring an endogenous nonsense mutation in the *TP53* gene[33]. After 24 h of treatment, proteins were extracted from the cells and the p53 protein was assessed by western blotting (Fig. 2a). After

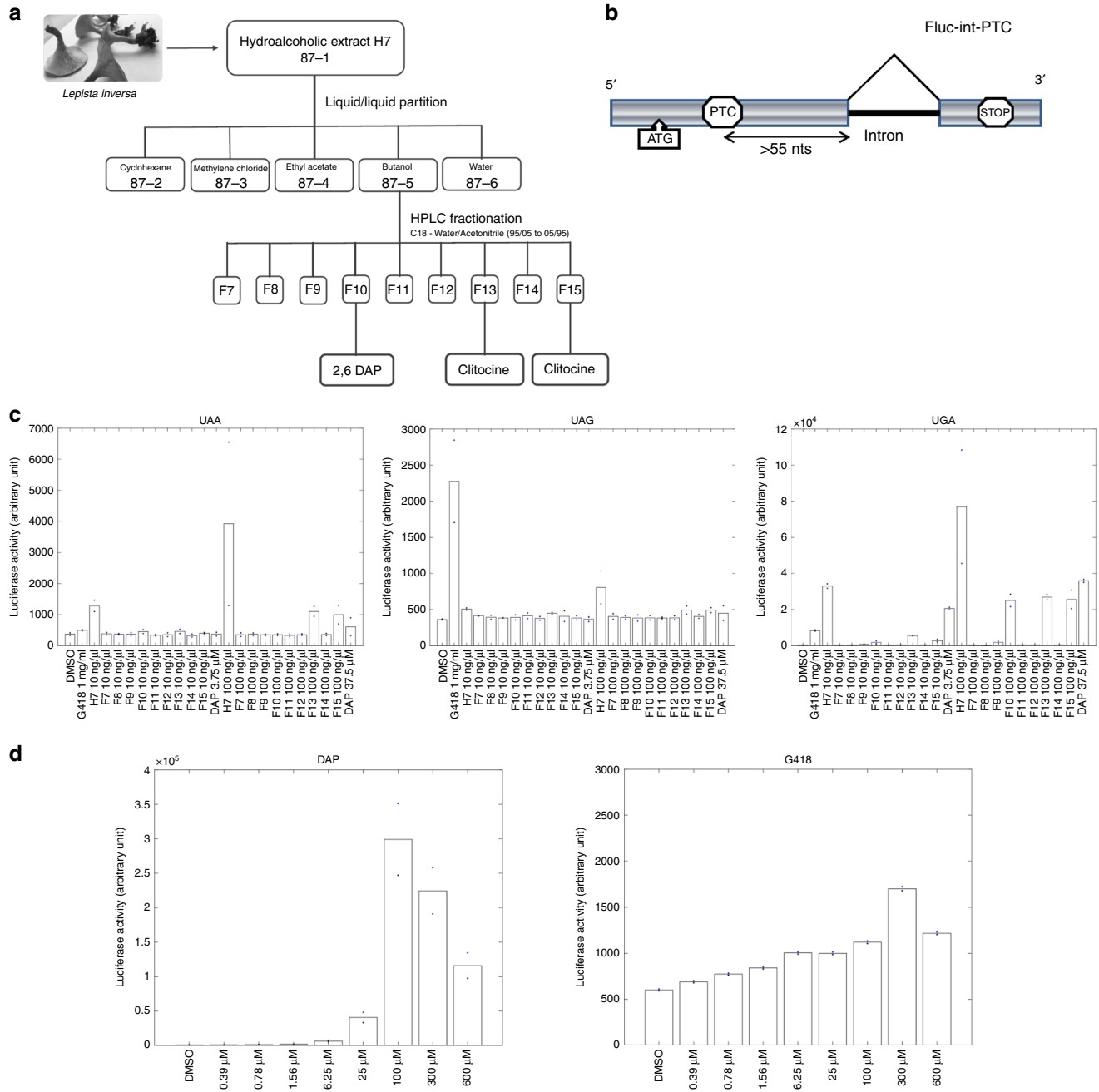

**Fig. 1 Identification of DAP as an efficient and exclusive corrector of UGA nonsense mutations. a** Fractionation pathway of H7 *L. inversa* extract, starting from fraction F87-1. **b** Schematic representation of the firefly luciferase construct used to measure readthrough activity. HeLa cells were transfected with an expression vector carrying a firefly luciferase (Fluc) gene consisting of the open-reading frame encoding firefly luciferase interrupted by an intron and a nonsense codon at position 109. **c** Luciferase activity in HeLa cells transfected with the construct described in **b** carrying a UGA, UAA, or UAG PTC and incubated with DMSO, H7 extract, one of various H7 fractions, or DAP. The value of each luciferase measurement is presented in the Source Data File. **d** Readthrough activity on the construct described in **b** as a function of the G418 or DAP dose. The value of each luciferase measurement is presented in the Source Data File. All data shown in **c**, **d** are representative of two independent experiments. Source data are provided as a Source Data file.

DMSO mock treatment, as anticipated, no p53 protein was detected, because of the presence of the PTC in p53 mRNA[33]. DAP treatment allowed dose-dependent synthesis of the p53 protein in Calu-6 cells (UGA nonsense mutation), but not in Caco-2 cells (UAG nonsense mutation) or Caov-3 cells (UAA nonsense mutation). This is consistent with the results in Fig. 1c and confirms that DAP exclusively corrects the UGA nonsense mutation. The p53 protein was detected after treatment with DAP at concentrations as low as 1.56 μM. It was barely detected upon treatment with 25 μM G418. This is consistent with the results in

Fig. 1c, showing that DAP corrects the UGA nonsense mutation more efficiently than G418.

Although DAP clearly promoted the synthesis of full-length proteins from UGA-nonsense-mutation-containing mRNAs, it was necessary to ascertain that the readthrough protein was functional. Evidence of this was first provided by the high luciferase activity observed after DAP treatment (Fig. 1c, d). To further demonstrate that DAP promotes the synthesis of functional full-length proteins from PTC-containing mRNAs, two additional assays were used. The first involved measuring the

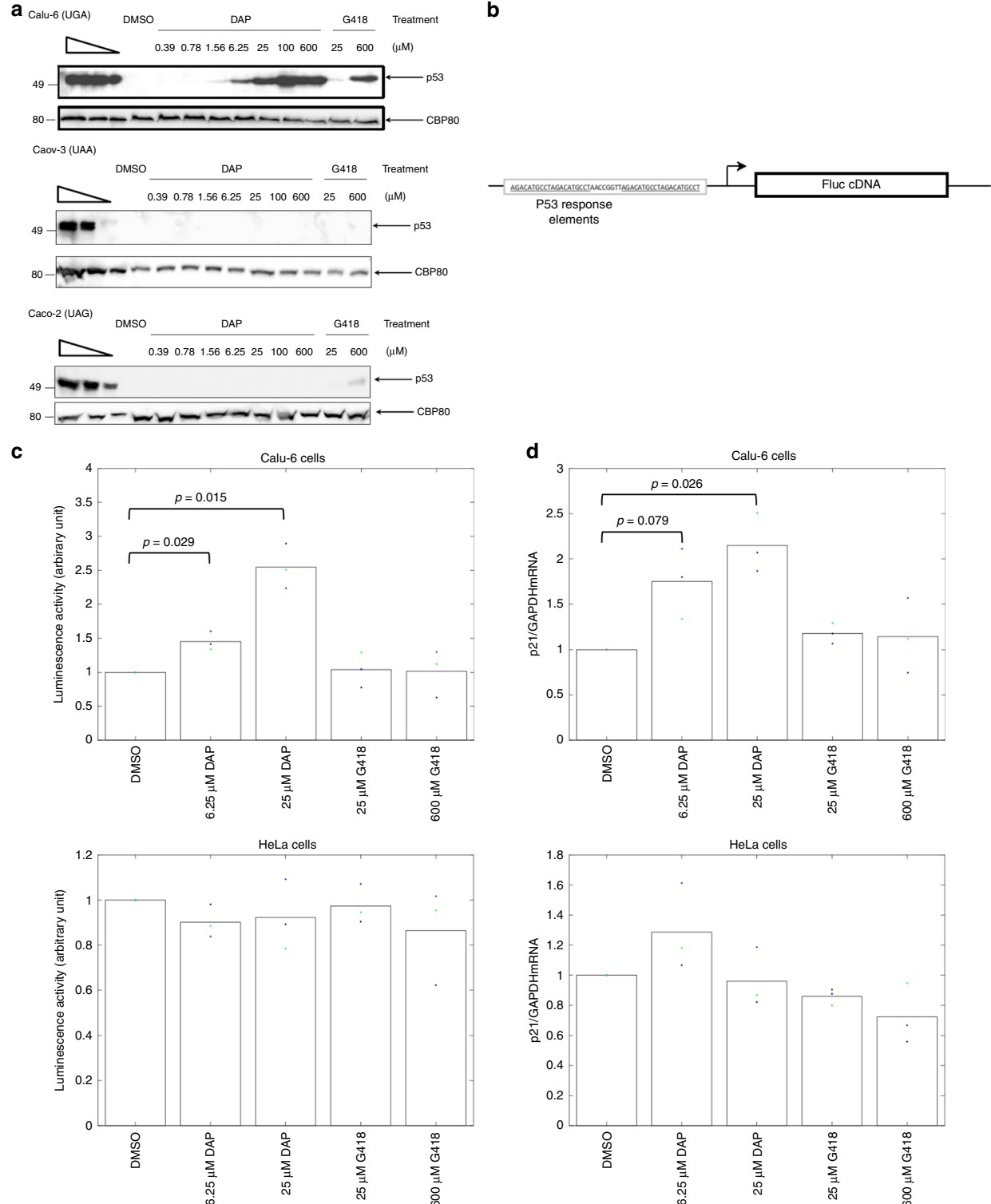

**Fig. 2 DAP restores the synthesis and function of p53 in Calu-6 cells. a** DAP restores expression of the endogenous *TP53* gene if it carries a UGA nonsense mutation but not if it carries a UAA or UAG nonsense mutation. Calu-6 (top), Caov-3 (middle), and Caco-2 (bottom) cells were incubated with increasing amounts of DAP or G418 before protein purification and monitoring of the p53 protein level by western blotting. The three leftmost lanes show twofold serial dilutions of untreated HeLa cell extract. The molecular weight is indicated to the left of each gel. **b** Schematic representation of the construct used to measure the function of p53. The construct carries a cDNA encoding the firefly luciferase under a promoter having two p53-responsive elements (underlined). The arrow indicates the transcription start. **c** Luciferase activity in Calu-6 (upper panel) and HeLa cells (lower panel) transfected with the construct described in **b** and treated with DMSO, DAP, or G418. The luciferase activity depends on the amount of functional p53 protein. **d** Level of p21 mRNA measured by quantitative RT-PCR in Calu-6 (upper panel) and HeLa cells (lower panel) treated with DMSO, DAP, or G418. The level of p21 mRNA is related to the amount of functional p53 protein synthesized in the cell. The results presented in panels **b** and **c** are representative of at least two independent experiments., *p*-values (*p*) were calculated using Student's *t*-test. Source data are provided as a Source Data file.

transcriptional activity of the p53 readthrough protein in Calu-6 cells transfected with an expression vector carrying the cDNA encoding firefly luciferase under the control of a promoter with two p53-responsive elements (Fig. 2b)[33]. Upon DMSO treatment, only basal luciferase activity was detected, reflecting promoter leakage (Fig. 2c). In the presence of increasing amounts of G418, the luciferase activity remained at its background level, indicating that G418 failed to promote synthesis of a functional full-length p53 protein. Upon DAP treatment, in contrast, the luciferase activity significantly increased, clearly demonstrating that the p53 readthrough protein synthesized in the DAP-treated Calu-6 cells was functional. That the p53 readthrough protein was directly responsible for the observed luciferase activity increase was confirmed by failure of DAP to cause a luciferase activity increase in HeLa cells expressing a wild-type *TP53* gene and transfected with the same luciferase construct (Fig. 2c, lower panel). The second assay used the p53 target gene p21 (ref. [44]). The level of endogenous p21 mRNA was monitored by quantitative RT-PCR in Calu-6 cells exposed to DMSO, G418, or DAP (Fig. 2d). It was found to increase significantly in cells exposed to DAP, but not in cells exposed to G418. When the same experiment was performed on HeLa cells, the level of p21 mRNA did not increase. This indicates that the effect of DAP observed in Calu-6 cells was due to increased *TP53* expression (Fig. 2d lower panel). This second assay thus confirmed that DAP promotes synthesis of a functional p53 protein in Calu-6 cells, more efficiently than does G418.

Overall, the results shown in Figs. 1 and 3 indicate that in all four tested models, DAP activates readthrough of UGA nonsense mutations to promote synthesis of functional read-through proteins.

**DAP does not inhibit NMD.** Correction of UGA nonsense mutations by DAP might result from NMD inhibition and read-through activity combined, as shown for other molecules[19,30], or from readthrough activation only. To check whether DAP inhibits NMD, the efficiency of NMD was monitored by measuring the level of p53 mRNA in the presence of DMSO, DAP, or G418 (Fig. 3a). Total RNA was purified from Calu-6 cells after 24 h of treatment. At both tested concentrations, DAP failed to inhibit NMD significantly. In contrast, G418 treatment caused a 1.8-fold increase in p53 mRNA. This is consistent with previous reports that G418 can both inhibit NMD and activate readthrough[19,30]. The fact that DAP does not affect NMD is also consistent with the observation that no inhibition of NMD was previously detected upon treatment with H7 extract[33].

**DAP shows low toxicity.** Although DAP-containing H7 extract does not appear to be toxic[33], the toxicity of DAP was assessed by measuring adenylate kinase activity in the culture medium of HeLa cells treated with DMSO, DAP, or the apoptosis inducer staurosporine (STS) used as a positive control (Fig. 3b). After 24 h of treatment, cells exposed to STS were found to have released adenylate kinase, whereas cells exposed to increasing amounts of DAP showed no release. This indicates that DAP is not cytotoxic, at least under these experimental conditions. This conclusion was supported by the results of two classic toxicity assays: measurement of propidium iodide incorporation into the DNA of dying cells and the MTT colorimetric assay assessing cell metabolic activity (Supplementary Fig. 4).

To strengthen the view that it is safe to use DAP, DNA integrity was assessed after DAP treatment. Briefly, mouse L5178Y TK$^{+/-}$ cells were exposed to DMSO or DAP for 24 or 3 h, followed by 24 h of recovery, in the absence and presence of Aroclor-induced rat liver S9 fraction. The idea was to reproduce the metabolizing conditions prevailing in the liver. Cells with

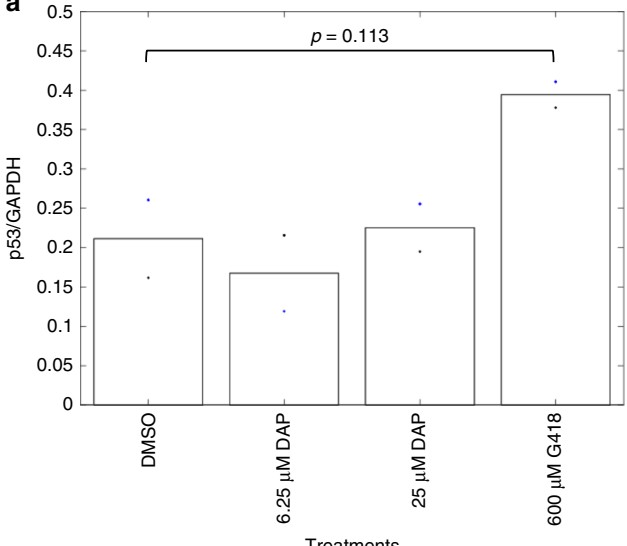

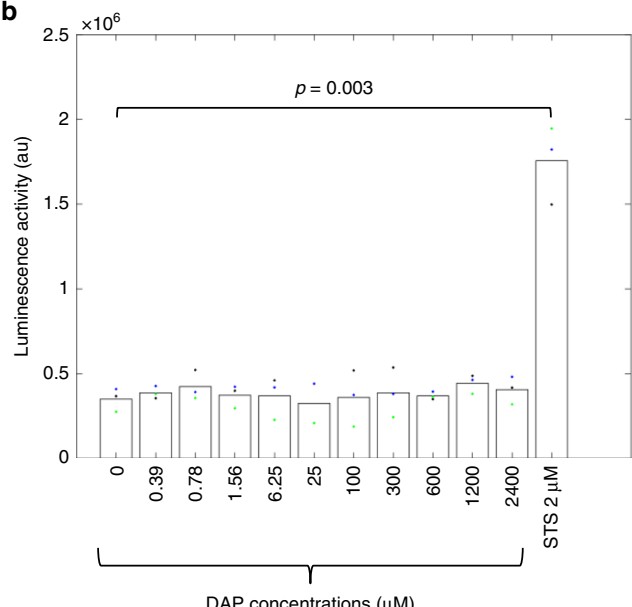

**Fig. 3 DAP is not an NMD inhibitor and is not toxic. a** Ratio of the level of p53 mRNA to that of GAPDH mRNA, as measured by RT-PCR, in Calu-6 cells treated with DMSO, DAP, or G418. **b** Adenylate kinase activity in the culture medium of HeLa cells treated with DMSO (0), increasing amounts of DAP, or the apoptosis inducer staurosporine (STS). The results presented in this figure are representative of two independent experiments. *p*-values (*p*) were calculated using Student's *t*-test. Source data are provided as a Source Data file.

nucleus injury, particularly ones with small nuclei, were then counted, these features being indicative of DNA damage and hence of test-compound genotoxicity (Table 1). No significant increase in the number of micronucleated cells was measured. Overall, these results demonstrate that DAP is not genotoxic.

To complete the study of possible DAP toxicity, we evaluated the consequences of DAP treatment, as compared to DMSO treatment, on gene expression at the transcriptional and translational levels. First, in HeLa cells, we used RNA-seq to assess the impact of DAP on the whole transcription profile. On the basis of statistics alone (adjusted *p*-value (Wald test) < 0.05), 3228 differentially expressed genes (DEGs) were detected (Supplementary Fig. 5). Among the DEGs, 775 showed at least

a two-fold change (adjusted *p*-value < 0.05, |log2FC| ≥ 1— Supplementary Table 4), 386 being downregulated and 389, upregulated. These 775 DEGs represent 3.38% of the total analyzed human genes (22,935 genes). Analysis of the DEGs revealed enrichment in Gene Ontology (GO) terms corresponding to the respiratory chain, electron transport, and oxidoreduction reactions (Supplementary Fig. 5D and Supplementary Data 1). Next, 2D differential gel electrophoresis (2D-DIGE)

was performed on the whole proteomes of HeLa cells exposed to DMSO or DAP. The two proteome profiles were found to overlap, without showing any spot shift at least not in substantial amounts (Pearson correlation coefficient: 0.94). This suggests that DAP does not promote synthesis of aberrant proteins (Supplementary Fig. 6). Altogether, these results indicate that DAP has a moderate impact on the gene expression program. The fact that it affects mainly energy-producing pathways could reflect a need for readthrough activation. The results strengthen the view, already supported by those of Fig. 4b and Supplementary Fig. 4, that DAP is not toxic to human cells.

**DAP increases in vivo readthrough of endogenous p53 UGA mutation**. The stability of DAP under simulated in vivo conditions was assessed in human liver S9 extract to mimic the liver environment. After incubation for 4 h, DAP and its metabolites were extracted with acetonitrile and subjected to HPLC/mass spectrometry analysis. No DAP-derived metabolites were detected under these experimental conditions, and recovery of the molecule after incubation exceeded 80%, indicating that DAP is stable and suggesting that it should be stable in vivo as well, since the liver is the main site of drug degradation in the body (Supplementary Fig. 7).

Next, the capacity of DAP to correct UGA nonsense mutations in vivo was assessed in a mouse model. Four-week-old immunodeficient nude mice were injected with an identical number of Calu-6 cells carrying a UGA nonsense mutation in the *TP53*

**Table 1 Micronucleus induction assay.**

| Treatment conditions | DAP dose (μM) | Micronucleated cells |
|---|---|---|
| Assay without S9 mix | 0 | 0 |
| | 20 | 2 |
| | 40 | 3 |
| | 70 | 1 |
| | Mitomycin (1 μg/ml) | 100 |
| Assay with S9 mix | 0 | 0 |
| | 40 | 0 |
| | 290 | 1 |
| | 580 | 0 |
| | Cyclopho sphamide (6 μg/ml) | 60 |

L5178Y TK$^{+/-}$ mouse lymphoma cells were exposed to increasing doses of DAP (as indicated), DMSO (DAP concentration 0 μM), or mitomycin or cyclophosphamide used as a positive control. 1000 cells were analyzed per condition and the number of micronucleated cells is reported. The experiment was performed in the absence (above) or presence (below) of S9 mix.

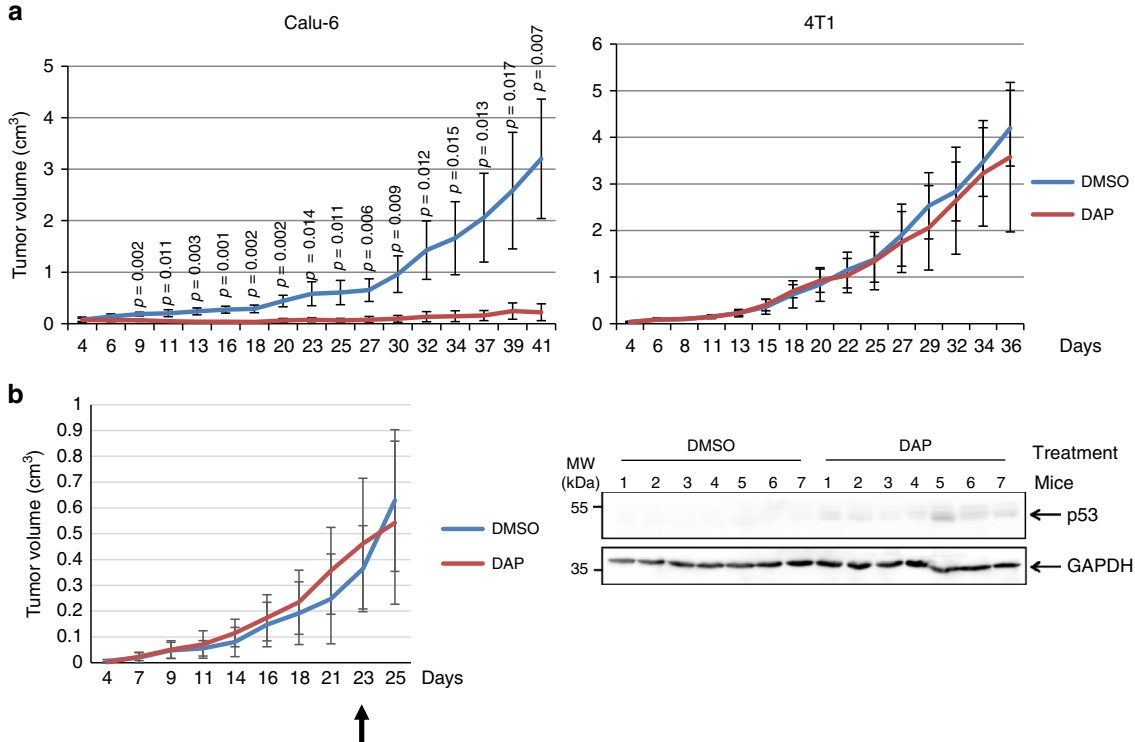

**Fig. 4 DAP corrects UGA nonsense mutations in vivo. a** Twenty-eight nude mice were injected with Calu-6 or 4T1 cells to promote tumor development. The mice were exposed to DMSO or DAP for about 5 weeks and tumor size was measured at different time after cell injection and plotted on the graph. For each experimental condition, 7 mice were used. **b** Fourteen nude mice were injected with Calu-6 cells to promote tumor development. The mice were exposed to DMSO or DAP when the average tumor size reached about 0.4 cm³. The arrow indicates the starting time of the exposure to DMSO or DAP. For each experimental condition, 7 mice were used. The tumor growth curve is presented on the left, and the right panel shows the western-blot analysis of p53 protein in the Calu-6-cell tumors of the seven DMSO-exposed and seven DAP-exposed mice. GAPDH was used as loading control. A molecular weight marker is indicated to the left of the gel. The results presented in this figure are representative of two independent experiments. *p*-values (*p*) were calculated using Student's *t*-test. Source data are provided as a Source Data file.

gene, of HCC-GR6 cells having a wild-type *TP53* gene or of p53-null 4T1 cells[45]. Under our experimental conditions, the tumors arising from 4T1 and Calu-6 cells in untreated mice showed similar growth rates. For 5 weeks, the mice were forced-fed every day with DMSO or DAP and tumor growth was monitored three times weekly. We assumed that if DAP promotes synthesis of a functional p53 protein in Calu-6 cells, these cells should divide more slowly and enter apoptosis as previously reported[24]. After 5 weeks of treatment, a significant difference was observed between Calu-6-cell-injected mice exposed to DMSO and ones exposed to DAP (Fig. 4a—left panel): DAP strongly impaired tumor growth as compared to DMSO. In contrast, when 4T1 or HCC-GR6 cells were used to induce tumor formation, no significant difference in tumor growth was observed between DAP-fed and DMSO-fed mice (Fig. 4a—right panel and Supplementary Fig. 8A). The slower growth of Calu-6-cell-derived tumors in DAP-treated mice is thus likely due to rescue of p53 protein synthesis in these tumors, as expected on the basis of our cell culture experiments and as observed after western-blotting applied to protein extracts of tumors from DAP-treated mice (Supplementary Fig. 8B). Because of the small size of the Calu-6 tumors in DAP-treated mice and thus the high probability of collecting mouse cells upon tumor excision, quantitation of p53 was done in another experiment. Mice were injected with Calu-6 cells to develop a tumor but were exposed to DMSO or DAP (for 1 week) only after the tumor size reached about 0.4 cm$^3$ (Fig. 4b). The p53 protein was detected in tumors isolated from DAP-fed but not DMSO-fed mice.

Altogether, these results suggest that in vivo, orally administered DAP rescues functional expression of genes carrying a UGA nonsense mutation.

**Molecular mechanism of DAP-driven UGA readthrough**. To elucidate the molecular mechanism of DAP-promoted UGA readthrough, mass spectrometry was used to identify the amino acid incorporated at the PTC position into firefly luciferase synthesized from the Fluc-int-UGA construct. HEK293FT cells treated with 25 μM DAP were chosen for their high rate of transfection, crucial to generating enough readthrough protein to be analyzed by mass spectrometry. The luciferase protein was digested with chymotrypsin to generate a PTC-position-containing peptide compatible with mass spectrometry analysis. Using this approach, we found only one amino acid, tryptophan, incorporated at the PTC position in the peptide of interest (Fig. 5a). Through non-Watson–Crick A*C base-pairing at the wobble position, human tRNA$^{Trp}$ has been shown to have suppressor functions[46], and plant tRNA$^{Trp}$ (CmCA) is also known to promote UGA readthrough[47].

To ensure accurate recognition, tRNA is subject to numerous posttranscriptional nucleotide modifications, notably in the anticodon triplet at wobble position $N_{34}$. Among the enzymes modifying tRNAs liable to recognize a stop codon by extended wobbling, human FTSJ1 (TRM7 in yeast) exclusively modifies tRNAs involved in UGA readthrough[12]. In addition to other tRNA targets, FTSJ1 methylates the cytosine at position 34 of tRNA$^{Trp}$. Inhibition of this modification could increase the capacity of tRNA$^{Trp}$ to misdecode and recognize the UGA stop codon. To test the hypothesis that DAP interferes with FTSJ1 activity, HeLa cells were co-transfected with the pFluc-int-UGA construct and with increasing amounts of an expression vector for either FTSJ1 or CTU1 (a uridine thiolase targeting U34 in tRNA$^{Gln}$ and tRNA$^{Lys}$, probably affecting recognition of the UAA stop codon)[48]. The transfected cells were then treated with increasing amounts of DAP (Fig. 5b). FTSJ1 and CTU1 overexpression were assessed by western-blot analysis (Fig. 5c).

Overexpression of FTSJ1, but not of CTUI, was found to strongly reduce DAP-promoted readthrough. This suggests that DAP promotes UGA readthrough by targeting FTSJ1 activity.

To determine whether DAP inhibits FTSJ1 activity, RiboMeth-Seq analysis was performed to detect modulation of tRNA 2′-O-methylation in the presence of DAP (Fig. 5d and Supplementary Fig. 9). HeLa cells were incubated with DMSO or DAP before extraction of small RNAs (which include tRNAs). RiboMethSeq analysis of human tRNAs was then performed[49]. In the presence of DAP, the level of 2′-O-methylation was significantly lower for three tRNAs. Notably affected was 2′-O-methylation of tRNA$^{Trp}$ at position 34 (Fig. 5d), while other known Nm positions were not affected (Supplementary Fig. 9). Although the human targets of FTSJ1 remain poorly characterized, modification of $Cm_{34}$ in tRNA$^{Trp}$ has been attributed to FTSJ1 (ref. [41]). It is noteworthy that a second affected tRNA (tRNA$^{Gln}$) also showed reduced $Cm_{32}$ methylation. This is consistent with DAP-dependent inhibition of FTSJ1 activity. The third hit was 2′-O-methylation of tRNA$^{Ser}$ at position $Gm_{18}$, attributed to TARBP1 activity. Overall, the results of the RiboMethSeq assay and the FTSJ1 overexpression experiment converge to suggest that DAP acts by inhibiting the action of FTSJ1 at certain 2′-O-methylation sites in human tRNAs.

Whether DAP interacts with FTSJ1 was then investigated by DAP affinity chromatography. The column packing was produced by covalent attachment of DAP to activated epoxy Sepharose beads. The column was then exposed to HeLa cell extract and this was followed by competitive elution of the bound proteins with excess DAP. The eluate was analyzed by western blotting for the presence of various proteins, including FTSJ1. Of all the tested targets, FTSJ1 was the only protein found to bind specifically to the DAP affinity column. It was absent in the eluate from a column packed with beads alone. These results indicate that DAP binds to FTSJ1 (Fig. 5e).

## Discussion

Nonsense mutations cause about 10% of genetic diseases[1] through silencing of the mutant gene. To date, we lack highly efficient molecules capable of rescuing the expression of genes harboring a nonsense mutation. Here we have focused on H7 extract from the common mushroom *Lepista inversa*, recently shown to promote expression of genes carrying UGA or UAA nonsense mutations[33]. We have identified two molecules responsible for the activity of H7, one being clitocine, previously shown to correct nonsense mutations more efficiently than G418 (ref. [24]). The mode of action of clitocine differs from that of other readthrough activator molecules, such as aminoglycosides, shown to interact with the ribosome[50]. Clitocine is incorporated into mRNA, where it replaces the adenosine base and facilitates stop codon recognition by some near-cognate tRNAs[24]. The second nonsense-mutation-correcting molecule found here in H7 extract is DAP. DAP corrects UGA nonsense mutations only; it is the first molecule to show correction of only one stop codon. The UGA stop codon is the most frequent termination codon, both physiological and premature[51]. Here, in three different assays, we show that DAP promotes synthesis of functional proteins, even though incorporation of a tryptophan at some positions is expected to be incompatible with protein function. DAP also corrects UGA nonsense mutations more efficiently than G418, even though it does not inhibit NMD. Although DAP appears more potent than G418, we were unable to calculate rescue of gene expression as compared to the wild-type as there exist no isogenic wild-type cells corresponding to the cell lines used here. Some other molecules such as PTC124/ataluren/Translarna[21] have been shown, like H7 extract[33], to promote PTC readthrough without affecting NMD. Remarkably, DAP does not appear toxic

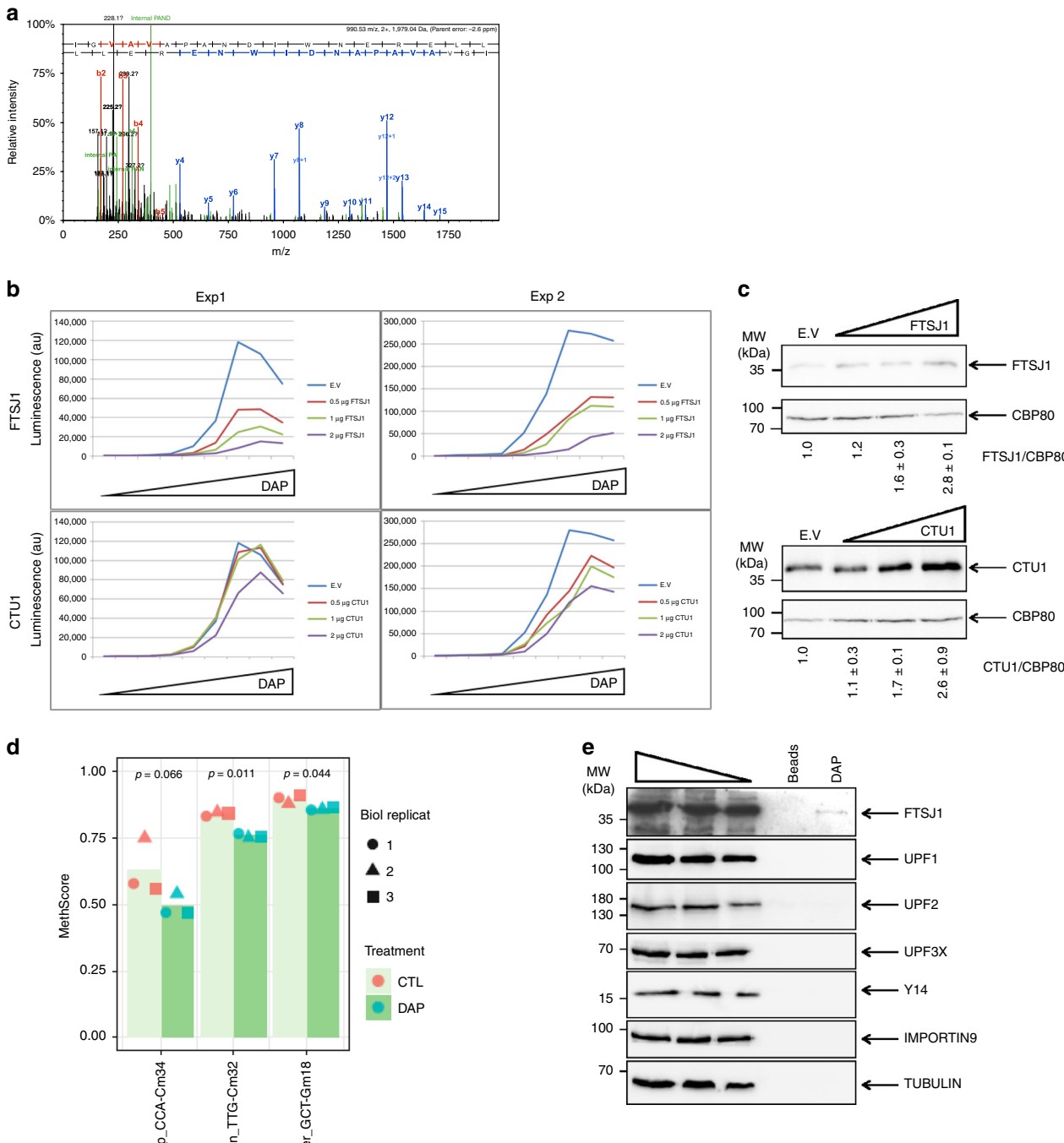

**Fig. 5 DAP interferes with the activity of FTSJ1. a** Determination by tandem mass spectrometry of the amino acid incorporated at the PTC position. Firefly luciferase from HEK293FT cells transfected with the Fluc-int-UGA construct was immunoprecipitated and digested with trypsin. The MS/MS spectrum of the peptide containing the PTC position and the sequence determination are shown. The upper red and blue sequences indicate the amino acids validated, respectively, by detection of Nter and Cter peptide fragments (b and y ion series). Peptide fragments y6 and y7 clearly identify tryptophan (W) at the PTC position. The black peaks correspond to non-attributed mass-to-charge ratio (m/z) values. The green peaks are internal fragments that do not contain an extremity of the peptide. **b** The efficiency of UGA readthrough by DAP decreases with increasing amounts of FTSJ1 but not with increasing amounts of CTU1. Readthrough efficiency was determined by measuring luciferase activity in HeLa cells co-transfected with the firefly luciferase construct described in Fig. 1b and with increasing amounts of an expression vector for either FTSJ1 or CTU1 (0.5, 1, or 2 μg). The empty expression vector (E.v.) was used as control. The cells were then exposed to 0, 0.39, 0.78, 1.56, 6.25, 25, 100, 300, or 600 μM DAP for 24 h before measuring the luciferase activity. The experiment was performed twice and the results of both experiments (Exp1 and Exp2) are shown. **c** Western blot analysis of FTSJ1 and CTU1. **d** 2′-O-methylation analysis of tRNA$^{Trp}$, tRNA$^{SER}$, and tRNA$^{GLN}$ by RiboMethSeq. HeLa cells were incubated for 24 h with DMSO or with 25 μM DAP. A MethScore was attributed to each tRNA 2′-O-methylation in tRNA purified from HeLa cells treated with DAP (dark green histogram) or DMSO (light green histogram). The results of three independent experiments are indicated with the symbols square, point, or triangle. **e** Analysis of DAP affinity column eluates shows that DAP interacts with FTSJ1. HeLa-cell extract was incubated with DAP covalently bound to beads or with beads alone. Proteins bound to the columns were eluted with excess DAP and the eluates analyzed by western blotting. A molecular weight marker is indicated to the left of each gel. p-values (p) were calculated using Student's t-test. Source data are provided as a Source Data file.

even at high concentration. Previously, DAP at 250 μM was shown to affect the cell cycle and to slow down cell growth strongly[52]. The cell type used in the cited work was the non-adherent mouse cell line L1210. In the present study, we have used adherent human cells. This might explain the difference in sensitivity observed between the two studies. In addition, to reduce the risk of possible DAP toxicity, 25 μM was the highest DAP concentration used in most of our experiments to characterize DAP readthrough. Hence, we may have underestimated the capacity of DAP to rescue the functions of genes carrying a UGA nonsense mutation (Fig. 2). Lastly, we show that in vivo orally administered DAP can rescue the expression of genes carrying UGA nonsense mutations.

As a purine derivative, DAP might promote PTC readthrough by replacing adenosines in RNA or DNA. Several lines of evidence, however, rule out this hypothesis. First, random exchange of adenosine by DAP would promote many false-sense mutations impacting many protein functions. In this study, the luciferase and p53 proteins synthesized after DAP treatment were demonstrated to be functional, and neither the treated cells nor the treated mice showed any sign of ill health. Second, DAP exclusively corrects UGA nonsense mutations. If adenosine exchange were its mechanism of action, one would expect DAP to correct all three stop codons, as adenosine is present in each of them. In fact, the mode of action of DAP appears original, since this molecule targets an enzyme responsible for the posttranscriptional modification of tRNAs. It inhibits the activity of the FTSJ1 methyltransferase, thus reducing Cm34 methylation in the tRNA$^{Trp}$ anticodon loop and thereby increasing readthrough at UGA stop codons only. tRNA modifications play essential roles in the fidelity of codon recognition, and their absence leads either to inefficient translation or to misdecoding through binding of near-cognate tRNAs. In our experiments, the only amino acid found inserted at the UGA stop codon was tryptophan. This indicates that the near-cognate tRNA$^{Trp}$ is responsible for the observed readthrough and that it misdecodes by forming a $C_{34}oA_{(+3)}$ at the third position of the codon (Trp is in the same codon box as the UGA stop). tRNA$^{Trp}$ is known to be 2′-O-methylated by FTSJ1 at position 34. Although structural knowledge is still lacking, all the existing evidence points to stabilization of a Watson–Crick pair at the third position, mediated by 2′-O-methylation leading to a $C_{m34} = G_{(+3)}$ Watson–Crick in the cognate complex. The absence of 2′-O-methylation is expected to allow misdecoding with recognition of the UGA stop codon through formation of a $C_{34}oA_{(+3)}$ pair with the near-cognate tRNA$^{Trp}$ (ref. [53]). Such a pair has been observed in the ribosomal structures of the Hirsch mutant[54] and involves a slight rotation of C34, leading to formation of a single H-bond pair between N4 (C34) and N1(A(+3)). The same pair between residue 34 and the third codon nucleotide was also observed in the ribosomal complex with tRNA$^{Ile}$, where C34 is modified at position 2, forcing C34 to re-orient slightly[55]. The enzymes responsible for tRNA modifications thus constitute a new set of putative targets for nonsense mutation correction, since their targeted inhibition should promote PTC readthrough. In addition, as posttranscriptional tRNA modifiers show specificity for particular tRNAs, positions, and modifications, their targeting might be adjusted to allow precise correction of nonsense mutations.

DAP is likely one of the most efficient UGA readthrough activators identified to date. Given its high efficacy and low toxicity, it seems more compatible with a therapeutic approach than previously reported molecules. Another advantage of DAP is its selectivity for the UGA stop codon, which should reduce the risk of potential side effects.

Correctors of nonsense mutations are possible drug candidates for the treatment of genetic diseases caused by nonsense mutations. They may also find a place in personalized medicine, since they would be used for patients with a particular mutation rather than for all patients with the same pathology. Only a fraction of patients suffering from a wide range of diseases should benefit from stimulated-readthrough therapy. The readthrough molecules identified so far are either highly toxic (e.g. aminoglycosides and clitocine) or not effective enough to offer sufficient therapeutic benefits (e.g. ataluren)[32,56–58]. New molecules are being developed to address all these issues, such as an ataluren derivative named PTC414, more effective than ataluren[59], and new-generation aminoglycosides[25,60,61], designed to increase the efficiency of readthrough correction and decrease the long-term toxicity associated with the previous generation. One of these new aminoglycosides, ELX-02, is currently being tested in a phase 1B clinical trial for cystic fibrosis (http://www.eloxxpharma.com/pipeline/).

In conclusion, several lines of evidence indicate that DAP is an excellent candidate drug for the treatment of genetic diseases caused by UGA nonsense mutations. Its correction efficiency is 10–200 times that of G418, suggesting that it could have therapeutic benefits. The results obtained in vivo in a mouse model suggest that DAP can rescue the functions of proteins synthesized from genes carrying nonsense mutations. In addition, it has been found to show low toxicity in cell cultures and no detectable toxicity in a mouse model. For all these reasons, DAP may someday offer new hope to patients whose pathology is caused by a UGA nonsense mutation.

## Methods

**Cell culture and treatment**. Human HeLa (ATCC; catalog number CCL-2), Caov-3 (ATCC; catalog number HTB-75), Caco-2 (ATCC; catalog number HTB-37) and mouse 4T1 cells (ATCC; catalog number CRL-2539) were grown in DMEM (Gibco) supplemented with 10% serum (Gibco) and 1% Zellshield. The cell lines Calu-6 (ATCC; catalog number HTB-56) and HCC827-GR6 (a generous gift from Pr. P.A. Jänne, Dana-Farber Cancer Institute, Boston, MA, USA)[62] were grown in RPMI 1640 (Gibco) supplemented with 10% serum (Gibco) and 1% Zellshield.

Cell transfections were performed with lipofectamine 3000 (Life technologies) according to the supplier's recommendations.

**Chemicals**. Clitocine and DAP used in all experiments were synthetic commercial samples from Azur Isotopes (Marseille, France; Ref. A-7052) and Sigma-Aldrich (St. Quentin-Fallavier, France; Ref. 247847), respectively. The commercial synthetic DAP sample was used as a reference for its identification in H7 extract and in all the biological assays described here.

**Constructs**. Expression vectors for FTSJ1/TRM7 (pEZ-V1624-M68-FTSJ1) and CTU1 (pEX-H4779-M02) were purchased from GeneCopoeia (Rockville, USA).

**Fractionation of Lepista inversa extract H7**. H7 extract was obtained from freeze-dried L. inversa mushrooms. These were crushed in a mortar and subjected to three successive extractions with methanol/water 50:50 v:v. Concentrating the H7 extract under reduced pressure (Rotavapor, Büchi) yielded fraction F87-1. Fraction F87-1 was solubilized in water and partitioned with organic solvents of increasing polarity (cyclohexane, methylene chloride, ethyl acetate, and butanol successively), yielding five fractions (F87-2–F87-5). The residual aqueous phase was named F87-6 (Fig. 1a). Fraction F87-5, a butanol phase of F87-1, was fractionated by semi-preparative high-performance liquid chromatography (HPLC) on a C18 Eclipse XDB column (21.2 × 150 mm, 5 μm; Agilent Technologies) coupled to a diode array detector (Agilent Technologies 1260 Infinity instrument). The mobile phases were (A) 95% water (0.05% trifluoroacetic acid), 5% acetonitrile and (B) 5% water (0.05% TFA), 95% acetonitrile. Separation was carried out at a flow rate of 10 mL/min, with the following gradient: 100% A for 6 min, 100% to 50% A in 30 min, 50% to 0% A in 2 min, 100% B for 2 min, 0–100% A in 2 min, and 100% A for 2 min. Compounds were detected at 210 and 254 nm. Nine fractions, numbered F7–F15, were obtained (Fig. 1a). From 40 mg of fraction F87-5, respectively 1.4, 2.4, and 1.6 mg of fractions F10 (Rt at 5.0 min), F13 (Rt at 7.0 min), and F15 (Rt at 9.5 min) were recovered. All organic solvents and reagents (acetonitrile, trifluoroacetic acid) were of HPLC grade purity (Ref. 83639.320 and Ref. 153112E, VWR, France). Ultrapure water obtained with a MilliQ water system (Millipore) was used for HPLC.

**Structure analysis by NMR and mass spectrometry**. NMR spectra were recorded at 298 K on either a 400 MHz Bruker Avance III HD or a 600 MHz Avance III

HD spectrometer equipped with a TCI cryoplatform (Bruker Biospin). Clitocine and DAP samples were dissolved in $D_2O$ or DMSO-d6 (Eurisotop, France) and placed in 5-mm Wilmad tubes (Eurisotop, France). $^{1}H$ and $^{13}C$ resonance assignments were carried out by means of series of conventional 1D $^{1}H$ and $^{13}C$ spectra and 2D $^{1}H$ COSY and $^{1}H$–$^{13}C$ HSQC and HMBC experiments performed under standard conditions.

ESI mass spectra were recorded with a hybrid ESI-Qq-TOF instrument equipped with an electrospray ionization source (API Q-Star Pulsar I, Applied Biosystems-Thermo Fisher Scientific or Compact, Bruker) operating in the positive ion detection mode. NMR and mass spectra were analyzed and compared with those obtained with authentic commercial synthetic samples of clitocine and DAP.

**Preparation of S9 mix**. The S9 mix was composed of 5 mM glucose-6-phosphate (Ref. G7879, Batch no. SLBP253V, Sigma, France), 33.5 mM KCl (Ref. P3911, Batch no. SLBQ3328V, Sigma, France), 8 mM $MgCl_2$ (Ref. M8266, Batch no. SLBK9099V, Sigma, France), 4 mM β-NADP (Ref. N5755, Batch no. SLBJ7935V, Sigma, France), and 100 mM $NaH_2PO_4$ (Ref. S3139, Batch no. BCBT3558, Sigma, France) in ultrapure water (Ref. BE51200, Batch no. BCBT3558, Lonza, France). Seventy-five microliters of human liver S9 fraction (Pooled Human Liver S9, 20 mg/ml, Ref. 452961, BD, France) was then added to 1380 μl of this S9 mix. The solution was warmed for 15 min at 37 °C in a thermostated water bath. The final protein concentration was 1 mg/ml after addition of working solutions.

**Protein extraction**. Proteins were extracted from $2 × 10^6$ cells in lysis buffer containing 5% SDS, 50 mM Tris–HCl pH 7, 20 mM EDTA, and a protease inhibitor cocktail (HALT Protease Inhibitor Cocktail, Pierce-Biotechnology, Rockford, IL, USA). Lysates were subjected to 30 pulses of sonication (Branson Digital Sonifier/amplitude 20%) before a 5 min centrifugation at 8000×g.

**Western blotting**. The equivalent of $2.5 × 10^5$ cells was subjected to 10% SDS–PAGE before transfer of the proteins to a nitrocellulose membrane. Membranes were incubated overnight at 4 °C in the presence of a 1/200 dilution of anti-p53 antibody (DO1) (Santa Cruz Biotechnology, Dallas, TX, USA; catolog number: sc-126) (Fig. 3a), a 1/2000 dilution of anti-p53 antibody (Biorad; catalog number: VMA00019) (Fig. 4) recognizing human and African green monkey p53, a 1/1000 dilution of anti-CBP80 (H-300) antibody (Santa Cruz Biotechnology, Dallas, TX, USA; catalog number: sc-48803), a 1/1000 dilution of anti-FTSJ1 antibody (Sigma-Aldrich, Saint-Louis, MO, USA; catalog number: AV49131), a 1/500 dilution of anti-CTU1 antibody (abcam, Cambridge, UK; catalog number: ab136083), a 1/1000 dilution of anti-UPF1 antibody (abcam, Cambridge, UK; catalog number: ab86057), a 1/500 dilution of anti-UPF2 antibody (generous gift from Pr. Lynne Maquat), a 1/1000 dilution of anti-UPF3X antibody (generous gift from Pr. Lynne Maquat), a 1/200 dilution of anti-Y14 antibody (Santa Cruz Biotechnology, Dallas, TX, USA; catolog number: sc-32312), a 1/1000 dilution of anti-TUBULIN antibody (abcam, Cambridge, UK; catalog number: ab134185), a 1/1000 dilution of anti-IMPORTIN9 antibody (abcam, Cambridge, UK; catalog number: ab52605), or a 1/1000 dilution of anti-GAPDH (6C5) antibody (Santa Cruz Biotechnology, Dallas, TX, USA; catalog number: sc-32233). After three washes of the membrane in TBS Tween, the membranes were exposed to a solution of peroxidase-coupled secondary antibody for detection of mouse-raised or rabbit-raised antibody (Jackson ImmunoResearch, Suffolk, UK) at a dilution of 1/5000 (catalog number: 115-035-003) or 1/50,000 (catalog number: 111-035-003), respectively. Antibodies were then detected with Super Signal West Femto Maximum Sensitivity Substrate (Pierce-Biotechnology, Rockford, IL, USA). Quantifications were performed with QuantityOne software (Biorad). The original un processed gel pictures are provided in the source data file.

**RT-PCR**. RNA was purified using RNazol (MRC) according to the manufacturer's instructions. 1/5 of the RNA preparation was used for reverse-transcription (RT) PCR (RT-PCR) using Superscript II (Life Technologies) for 2 h at 42 °C in the presence of random hexamers. The resulting cDNAs were PCR amplified in the presence of dCTP(α$^{33}$P) (Perkin Elmer) with the following primers: GAPDH mRNA (sense: 5′-CATTGACCTCAACTACATGG-3′; antisense: 5′-GCCATGC CAGTGAGCTTCC-3′), p53 mRNA (sense: 5′-ATGTGCTCAAGACTGGCGC-3′; antisense: 5′-GACAGCATCAAATCATCC-3′), p21 mRNA (sense: 5′-CGAGGCA CTCAGAGGAG-3′; antisense: 5′-TCCAGGACTGCAGGCTTCC-3′). PCR products were quantified with Personal Molecular Imager and QuantityOne quantification software (Bio-Rad).

**Firefly luciferase assay**. HeLa cells were transfected with the FLuc-int-UGA construct using lipofectamine 3000 (Life technologies) according to the manufacturer's recommendations. The cells were distributed 24 h later into white 96-well plates (Corning, Corning, NY, USA) and incubated for 20 h at 37 °C and 5% $CO_2$ with DMSO, DAP, or G418 at 1 mg/ml. Steadylite plus luciferase substrate (Perkin Elmer, Waltham, MA, USA) was then added to each well before measuring the visible light from each well for 10 s with a Tristar luminometer (Berthold technologies, Versailles, France). The plate was read twice and the reported values are averages of the two reads.

**Micronucleus induction assay**. The assay was performed by CytoxLab (Evreux, France). Briefly, L5178 TK$^{+/−}$ mouse cells were exposed to three concentrations of DAP for 24 or 3 h. This was followed by a 24-h recovery period when the experiment was performed in the presence of Aroclor-induced rat liver S9 metabolizing mix (at 2% final concentration). As a positive control, mitomycin C was used at 1 μg/ml in experiments without S9 mix and cyclophosphamide was used at 6 μg/ml in experiments with S9 mix. The nuclei were then stained with Giemsa before counting the micronuclei among 1000 mononucleated cells.

**Toxicity assay**. The toxicity assay was performed 24 h after starting cell treatment. The cell culture supernatant was centrifuged at 6000×g for 10 min to pellet cells and cell debris. One-twentieth of the supernatant was then used to measure adenylate kinase activity according to the supplier's protocol (Lonza Biosciences, Basel, Switzerland).

**In vivo assay**. To initiate tumor growth, 6-week-old nude mice received subcutaneously 10 million cells suspended in PBS. Starting 24 h after injection, the mice were force-fed daily with 200 μl DMSO or DAP solution at 5 mg/ml. Tumors were measured three times weekly for 5 weeks to determine the tumor volume by the following formula: $l$ (mm) × $L$ (mm) × $L$ (mm). The tumors were then collected and a fraction was used for western blot analysis after protein extraction. All in vivo experiments complied with all relevant ethical regulations for animal testing and research and were authorized by the Ethic Committee 075 under the number 12971-2018010815336092.

**tRNA modification analysis**. tRNAs were purified with RNAzol (MRC, Cincinnati, OH, USA) from HeLa cells exposed for 24 h to DMSO or 25 μM DAP, according to the manufacturer's protocol. The detailed protocol and a description of the RiboMethSeq assay are provided elsewhere[49].

**DAP affinity columns**. Two hundred milligrams of Epoxy-activated Sepharose 6B (GE Healthcare) was suspended in distilled water and washed with 80 ml distilled water and then divided into two samples. Coupling was performed by adding 100 μmol DAP suspended in 10 N NaOH to the first sample and incubating for 16 h at 37 °C under gentle stirring. The packing for the "empty column" was obtained by adding 1 ml of 10 N NaOH to the second sample and proceeding similarly. The columns were then washed with 10 N NaOH and the remaining active groups were blocked by incubating the columns overnight at 37 °C and with ethanolamine, pH 8. The columns were further washed with three cycles of 0.1 M $CH_3COOK$ pH 4 containing 0.5 M NaCl, followed by 0.1 M Tris–HCl pH 8 containing 0.5 M NaCl. Each column was then incubated overnight at 4 °C under gentle stirring with an amount of extract equivalent to 120 million cells, in the presence of HALT Protease Inhibitor Cocktail (Pierce). The columns were then washed five times with a buffer containing 50 mM Tris–HCl pH 7.4, 300 mM NaCl, and 0.05% NP40, before elution with 100 μl of 100 mM DAP dissolved in DMSO.

**Amino-acid identification assay**. Sixty million HEK293FT cells were transfected with 120 μg Fluc-int-UGA construct with $CaCl_2$ in HBSS buffer. After a 24-h incubation at 37 °C under 5% $CO_2$, the cells were exposed for 24 h to 25 μM DAP or to DMSO as negative control, before immunoprecipitation of firefly luciferase with anti-firefly luciferase antibody (EDM Millipore, Temecula, USA; catalog number: AB3256) according to the protocol described in ref. [63]. The immunoprecipitate was subjected to 10% SDS–PAGE and then firefly luciferase was detected by Coomassie staining. The band at 64 kDa present in DAP-treated cells but not in cells exposed to DMSO was excised for in-gel digestion with chymotrypsin. NanoLC–MSMS analysis of the protein digest was performed on an UltiMate-3000 RSLCnano System coupled to a Q-Exactive instrument (Thermo Fisher Scientific).

MS/MS data were interpreted with the Mascot search engine (version 2.4.0, Matrix Science, London, UK) installed on a local server. Searches were performed with a tolerance on mass measurement of 10 ppm for precursor ions and 0.02 Da for fragment ions, against a database built with Fluc sequences degenerated at Tyr109, recombinant trypsin, and a list of classical contaminants (138 entries). Cysteine carbamidomethylation, methionine oxidation, protein N-terminal acetylation, and cysteine propionamidation were searched for as variable modifications. Up to three missed chymotrypsin cleavages were allowed.

**DAP stability assay**. The assay was performed by C-Ris Pharma, Saint Malo, France. Briefly, DAP at 10 mM was diluted in S9 mix (to 300 μM final concentration) and acetonitrile extraction was performed either immediately (T0) or after a 4-h incubation at 37 °C. The samples were then centrifuged and directly analyzed by LC–MS. The extraction efficiency was estimated at 97.2%. After extraction, the collected supernatants were directly injected into the HPLC system for analysis. Analysis was performed by MS to allow detection of potential metabolite(s).

**Reporting summary**. Further information on research design is available in the Nature Research Reporting Summary linked to this article.

## Data availability

A reporting summary for this Article is available as a Supplementary Information file. The source data underlying Figs. 1c, d, 2a, c, d, 3, 4, and 5 and Supplementary Figs. 4, 8, and 9 are provided as a Source Data file. Accession numbers for the RiboMetSeq and RNA-seq data are PRJEB36039 and PRJEB36055, respectively, from the following URL: https://www.ebi.ac.uk/ena. All data is available from the corresponding author upon reasonable request.

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

## Acknowledgements

Authors would like to thank Dr. Jens Lykke-Andersen, Pr. P.A. Jänne and Pr. Lynne Maquat for reagents, Dr. Clément Carré, Dr. Bruno Lapeyre, Dr. Gabriele Neu-Yilik, Dr. Matthias Hentze, and Dr. Jean-Paul Renaud for helpful discussions. Authors also thank the Bicel facility for technical help, the PLEHTA for technical help on mouse managing and in vivo experiments, Valérie IGEL-BOURGUIGNON and Dr. Lilia AYADI from the Next-Generation Sequencing Core Facility of UMS2008 IBSLor (Université de Lorraine-CNRS-INSERM) for their help in RiboMethSeq library preparation and sequencing. F.L. is an Inserm researcher supported by fundings from Vaincre la mucoviscidose, the Association française contre les myopathies, the GIP Cancéropôle Nord Ouest, the Fondation Maladies Rares and the SATT Lutech.

## Author contributions

C.T., S.A., C.B., C.L., V.M., J.-M.S., H.B., T.C., R.G., D.H., A.M., M.-C.C., P.-A.M., and F.L. performed experiments. S.A., C.B., V.M., E.D.-B., J.-M.S., D.T., Y.M., S.R., and F.L. wrote the manuscript. C.T., S.A., C.B., E.D.-B., J.-M.S., E.Wer., D.H., E.A., A.K., E.Wes., Y.M., S.R., and F.L. designed the experiments and analyzed the data.

## Competing interests

The authors declare no competing interests.
