## [Peer Review File · Nature Communications]

Reviewers' comments:

Reviewer #1 (Remarks to the Author):

The identification and characterization of small molecules that promote the readthrough of premature termination codons created by nonsense mutations is an important area of study. Such small molecules could be very useful to treat patients with cancers or inherited disorders caused by nonsense mutations. The search for chemicals that can induce sufficiently high levels of readthrough to ameliorate disease, while being sufficiently non-toxic to be used in chronic treatment, has been frustratingly difficult and not highly successful. Some aminoglycosides and derivatives can induce high levels of readthrough but they are significantly toxic, while the non-toxic readthrough compounds described to date show low or even unconvincing readthrough activity.

The discovery that 2,6-diaminopurine (DAP) can induce high levels of readthrough at TGA nonsense mutations at non-toxic concentrations is important and interesting. Moreover, the authors' identification of the FTSJ1 tRNA 2'-O-methyltransferase as a potential target is interesting and original.

The manuscript has some very significant strengths:

- The levels of p53 readthrough induced by DAP in Calu-6 cells with R196X (TGA) nonsense mutation are high and they are observed at relatively low DAP concentrations. In this assay, DAP is a more potent and efficacious readthrough inducer than G418, arguably the benchmark compound in the field.
- The experiments describing the mechanism of action are interesting and together constitute a fairly convincing package: 1- Mass spectrometry identified tryptophan as the only amino acid incorporated at the PTC; 2- This observation led to the hypothesis that FTSJ1 function might be inhibited by DAP. Consistent with this hypothesis, overexpression of FTSJ1 antagonized DAP-induced readthrough; 3- RiboMethSeq analysis showed that the 2'-O-methylation of tRNA^{trp} was reduced upon incubation of cells with DAP; and 4- FTSJ1 was found to bind to a DAP affinity column.

The manuscript also shows very significant weaknesses:

- The use of an adenylate kinase release assay to measure cell toxicity (Fig. 4B) is perplexing given the fact that DAP is an adenine analog. DAP present in the cell culture medium could well inhibit any adenylate kinase released from the cells. The adenylate kinase assay is not widely used in the literature. The data would be more convincing if a different, more widely used cytotoxicity assay were used instead. This is an important point because an important claim of this study is that DAP shows low cell toxicity at concentrations that induce readthrough and DAP has previously been shown to be cytotoxic to other cell lines within that concentration range, e.g. L1210 leukemia cells (Weckbecker, *Adv Enzyme Regul.* 28:125, 1989), a point that is not discussed.

- The effect of DAP on Calu-6 tumor growth in vivo (Fig. 5B) is impressive but it does not constitute conclusive evidence that DAP can correct nonsense mutations in vivo. DAP has been characterized as an antimetabolite of purine nucleotide metabolism that can inhibit DNA and RNA synthesis and inhibit the proliferation of cancer cell lines without p53 nonsense mutations, as stated above (Weckbecker, *Adv Enzyme Regul.* 28:125, 1989). Therefore, DAP could inhibit tumor growth by a mechanism that has nothing to do with p53 readthrough. Additionally, the small levels of p53 detected by western blotting in DAP-treated tumors (Fig. 5C) is not conclusive evidence of readthrough. This p53 could well come from mouse cells present in the residual tumor mass rather than from readthrough in Calu-6 cells. Indeed, the tumors were collected after three weeks of treatment, when tumor size was barely detectable and could well consist of only mouse cells. Moreover, the DO-1 antibody used is documented to recognize both human and mouse p53. A more convincing experiment would be to let the tumors grow to a sizeable volume to ensure that most cells are Calu-6 and then treat mice with DAP for a few days before sacrificing and measuring p53.

- Figure 3C showing a 2-fold increase in p21 mRNA by 25 μ M DAP is weak evidence that p53 produced by readthrough is functional. As a comparison, Bidou et al (*RNA Biol.* 14:378, 2017) reported that the expression of p21 “was strongly induced by NB124 treatment (induction factor of 150)”; (NB124 is a readthrough aminoglycoside).

- Although the proposed mechanism involving FTSJ1 is attractive, the authors do not cite previous papers showing that DAP is an adenine analog that can be incorporated into nucleotides (DAP riboside 5'-triphosphate). Therefore, alternative mechanisms of action including incorporation of DAP into tRNA or mRNA and interference with the multiple cellular reactions that utilize ATP should at least be discussed.

Additional points:

- The manuscript is generally written clearly but it needs English language improvement.

- Page 5: the statement that the development of PTC124 was terminated is incorrect. It is not FDA-approved but it is approved for the treatment of nmDMD in the EU.

- Page 5: the statement that DAP has been injected into children without showing any sign of toxicity (Ref. 36, Burchenal) should be qualified. This reference states that “the patients were instructed to spread the oral dose over the whole day in an attempt to decrease the nausea that frequently accompanies the administration of the drug”.
- Zellshield (page 6) is a mixture of antibiotics. The authors should ascertain whether it contains G418 or aminoglycosides that might affect readthrough.
- The NMR spectra and other data permitting the identification of the isolated compound as DAP should be provided in appendix. Has the structure of the commercial DAP been ascertained independently?
- It should be stated whether all experiments beyond Fig. 1C were carried out with commercial DAP (as opposed to isolated compound).
- Page 14: to compare the potencies of DAP and G418, their EC50 values should be determined and compared rather than providing statements such as “at 6.25 μ M, it was 10 times as effective as 600 μ M G418”.
- Figure 3B showing that DAP increases expression of the Fluc reporter by 2.5-fold over DMSO is weak evidence that p53 produced by readthrough is functional.
- I am not qualified to evaluate the mass spectrometry data presented in Fig. 6A but the panel seems difficult to interpret.
- In the RiboMethSeq experiment presented in Fig. 6C, it should be indicated what DAP concentration was used and how long the cells were incubated for.
- Figure 5B showing no DAP modification by S9 liver extract is unclear and the figure is insufficiently labeled. This panel should be moved to supplemental data.
- The last sentence on page 19 (Inhibition of this modification...) should be supported by a reference.

Additional note: This reviewer does not have the background required to expertly review the data presented in Figures 3D and 6A.

Reviewer #2 (Remarks to the Author):

The aim of this manuscript is to characterize the effect of DAP on readthrough of UGA. The authors conclude that DAP corrects the UGA nonsense mutation with high efficiency in cell lines, in vivo in mouse models and in patient cells.

There are several major concerns that limit the ability to estimate the efficiency of DAP as a novel read-through agent:

1. The functional assays are lacking WT controls as a reference. Without comparison to the WT function, the efficiency and level of correction cannot be concluded. This should be performed for the p53 and the CFTR analyses. Only in the mice experiments (Fig 5) a WT p53 control level is shown.
2. As was shown in the past, each cell type/line has a different NMD efficiency, which affects the level of read-through. The effect of NMD efficiency in each of the studied cellular systems is missing. The authors should include experiments with either NMD inhibitors or siRNA against key NMD factors.
3. The authors conclude that DAP has low toxicity. This is based on two assays. One is the effect of DAP on gene expression. The authors conclude that only 0.13% of the genes showed a change greater than 4 folds. The experiment was performed in HeLa cells. A previous work in HeLa cells by Mendell JT et al., 2004 in Nature Genetics identified ~200 transcripts that are physiologically regulated by NMD. The genes in this group (those that are carrying UGA premature stop) should have been affected by DAP. Moreover, the authors chose a threshold of 4X as a change in expression. This is a too high rough as lower changes may have a profound physiologic effect, as shown in Mendell's paper. I could not find the list of genes that were affected by DAP. It is important to have this list. The authors should present the list of genes that were changes by at least 2 fold.

The other assay is measuring AK activity. A more thorough study of cell toxicity should be performed. Assessing cell membrane integrity is one of the most common ways to measure cytotoxic effects. MTT or SRB are other commonly used assays.

4. The in vivo experiments in mice presented in Figure 5B are not convincing. The control cells are inadequate, since their tumor development is very different from that in Calu 6 cells with nonsense p53 mutation, as can be seen in the DMSO results. The authors should have chosen cells with the same tumor growth rate. The concern is that DAP may have an anti-proliferative effect and not a specific effect for UGA nonsense mutations.

5. As was shown in the past, each cell type/line has a different NMD efficiency, affecting the level of read-through. The effect of NMD efficiency in each of the studied system is missing. The authors should include experiments with either NMD inhibitors or siRNA against key NMD factors.

Other comments:

1. Page 17 4 lines from the bottom: what does this mean "no visible differences"? is this an adequate way of analyzing the result?

2. Page 18 – the title is misleading since the assay is ex vivo and not in vivo. The ref no 49 is not relevant.
3. CFTR functional assay – CFTR function in nasal epithelial cells from patients should have been the Ussing Chamber using differentiated cells. This is the standard functional assay to test the effect of drugs on CFTR function. The assay used by the authors, is the SPQ assay. Where the cells differentiated? No details on the analyses is given and reference is to a review paper.
4. Figure 2 – the first 3 lanes in each of the panels are unclear. What is the identity of the experiment performed in these three lanes?
5. The results presented in Figure 3B and C are inconsistent with the results presented in Figure 2. This includes the G418 different concentrations as well as DAP in comparison to G418. For example, in Figure 3B and C, there is no difference in the effect of G418 using 25 or 600 μ M whereas in the Western blot presented in Figure 2, there is a significant difference in p53 expression between these concentrations. More importantly, 25 μ M DAP has the same effect on the p53 protein level as 600 μ M G418 whereas in Figure 3B and C the effect of these two treatments is very different. These results are confusing. Can the authors explain them?
6. Figure 5C – at what time point was the analyses performed?
What is the 1-7? Different mice? Different days?
7. Figure 5C shows a very low level of WT p53 in the DAP experiment. This may indicate that the tumor development results are due to a combined effect of DAP on UGA stop and its activity as an antiproliferative agent.
8. Figure 5 – the numbers in X axis should be explained. Are these time points of analysis? There were 3 analysis a week for 21 days (according to the text). What are the 17 data points?
9. Proteome analysis – the authors should perform the proteomic analysis on Calu6 and not on HeLa. In order to be able to identify the change in p53.
10. The manuscript would benefit from English editing

Reviewer #3 (Remarks to the Author):

This reviewer thought this paper was very interesting and presents new important information on DAP activity. The evidence is mostly convincing and DAP seems like a promising molecule for development of genetic disease treatment caused by nonsense mutation. There are a few questions and suggestions I have regarding parts of the paper and experiments.

First, will DAP cause readthrough of normal UGA stop codons? If it does, it will be toxic and not desirable for the organism. If not, how does it distinguish between UGA nonsense mutation and normal UGA at the end of the ORF?

Another question pertains the effect of DAP on healthy mice. In figure 5B, the effect of DAP on tumor growth in immunodeficient mice was shown. What, if any, would be the effect of DAP on healthy mice?

In figure 6C, can you claim significance for 2'-O-methylation of tRNA-Trp in the presence of DAP, since $p=0.066$? Would this cast slight doubt into the effect of DAP in inhibiting methylation of tRNA by FTSJ1?

Further, to strengthen this important point of DAP inhibiting FTSJ1 activity, an experiment to determine whether the inhibition is competitive or non-competitive with regard to the methyl donor/tRNA and calculating the K_i value for DAP would be useful.

Regarding figure 6B, the text does not describe the western blot data in the right panel. The point of that data is a little unclear to this reviewer.

Responses to the reviewer concerns

We would like to thank the reviewers for their comments and suggestion that bring significant strength in the revised version of the manuscript.

Reviewer 1

- The use of an adenylate kinase release assay to measure cell toxicity (Fig. 4B) is perplexing given the fact that DAP is an adenine analog. DAP present in the cell culture medium could well inhibit any adenylate kinase released from the cells. The adenylate kinase assay is not widely used in the literature. The data would be more convincing if a different, more widely used cytotoxicity assay were used instead. This is an important point because an important claim of this study is that DAP shows low cell toxicity at concentrations that induce readthrough and DAP has previously been shown to be cytotoxic to other cell lines within that concentration range, e.g. L1210 leukemia cells (Weckbecker, *Adv Enzyme Regul.* 28:125, 1989), a point that is not discussed.

R: The reviewer is right about the results from the referenced paper. We have now performed two additional, standard assays to measure DAP toxicity: a propidium iodide incorporation assay and the MTT assay (new Fig. S4). With neither assay do we detect any toxicity at any tested concentration. The difference with the Weckbecker study might lie in the cell type used. We have added the following sentence in the discussion to raise this interesting point: **Previously, DAP at 250 μ M was shown to affect the cell cycle and to slow down cell growth strongly⁵⁴. The cell type used in the cited work was the mouse non-adherent cell line L1210, whereas we, in the present study, have used human adherent cells. This might explain the difference in sensitivity observed between the two studies.**

- The effect of DAP on Calu-6 tumor growth in vivo (Fig. 5B) is impressive but it does not constitute conclusive evidence that DAP can correct nonsense mutations in vivo. DAP has been characterized as an antimetabolite of purine nucleotide metabolism that can inhibit DNA and RNA synthesis and inhibit the proliferation of cancer cell lines without p53 nonsense mutations, as stated above (Weckbecker, *Adv Enzyme Regul.* 28:125, 1989). Therefore, DAP could inhibit tumor growth by a mechanism that has nothing to do with p53 readthrough. Additionally, the small levels of p53 detected by western blotting in DAP-treated tumors (Fig. 5C) is not conclusive evidence of readthrough. This p53 could well come from mouse cells present in the residual tumor mass rather than from readthrough in Calu-6 cells. Indeed, the tumors were collected after three weeks of treatment, when tumor size was barely detectable and could well consist of only mouse cells. Moreover, the DO-1 antibody used is documented to recognize both human and mouse p53. A more convincing experiment would be to let the tumors grow to a sizeable volume to ensure that most cells are Calu-6 and then treat mice with DAP for a few days before sacrificing and measuring p53.

R: To meet the reviewer's requirement, we have removed the original p53 western blot and performed the experiment suggested by the reviewer. We now show the new p53 western blot for tumors about 0.6 cm³ in size, excluding the possibility of having mainly collected mouse cells

(Figure 5B). In addition, we have used another antibody that recognizes human and African green monkey p53 only, and not the mouse p53 protein (reference VMA00019 from Bio-Rad).

- Figure 3C showing a 2-fold increase in p21 mRNA by 25 μ M DAP is weak evidence that p53 produced by readthrough is functional. As a comparison, Bidou et al (RNA Biol. 14:378, 2017) reported that the expression of p21 “was strongly induced by NB124 treatment (induction factor of 150)”; (NB124 is a readthrough aminoglycoside).

R: The difference in results can be explained by the experimental conditions. In the study of Bidou et al., HDQ-P1 cells were used and not Calu-6 cells (used in our study). The cell line used in the study can strongly influence the intensity of the response to a drug, and so can the treatment duration (72 hours for the Bidou study and 24 hours for our study). In addition, the mutation in p53 is not at the same position and could also influence the reactivity of the readthrough protein.

- Although the proposed mechanism involving FTSJ1 is attractive, the authors do not cite previous papers showing that DAP is an adenine analog that can be incorporated into nucleotides (DAP riboside 5'-triphosphate). Therefore, alternative mechanisms of action including incorporation of DAP into tRNA or mRNA and interference with the multiple cellular reactions that utilize ATP should at least be discussed.

R: We have now added the following text to discuss this specific point:

As a purine derivative, DAP might promote PTC readthrough by replacing adenosines in RNA or DNA. Several lines of evidence, however, rule out this hypothesis. First, random exchange of adenosine by DAP would promote many false-sense mutations impacting many protein functions. In this report, the luciferase, p53, and CFTR proteins synthesized after DAP treatment are demonstrated to be functional, and neither the treated cells nor the treated mice showed any sign of massive metabolic malfunction. Second, DAP exclusively corrects UGA nonsense mutations. If adenosine exchange were its mechanism of action, one would expect DAP to correct all three stop codons, as adenosine is present in each of them.

Additional points:

- The manuscript is generally written clearly but it needs English language improvement.

R: The manuscript has been corrected by a professional English corrector, who is also a scientist.

- Page 5: the statement that the development of PTC124 was terminated is incorrect. It is not FDA-approved but it is approved for the treatment of nmDMD in the EU.

R: The reviewer is right and we have changed the sentence as follows:

As the results, albeit encouraging, were too limited to offer patients a significant benefit, **this molecule has not been approved in the United States for the treatment of nonsense-**

mutation-related Duchenne muscular dystrophy when it has received a marketing authorization in Europe for this pathology^{31, 32}.

- Page 5: the statement that DAP has been injected into children without showing any sign of toxicity (Ref. 36, Burchenal) should be qualified. This reference states that “the patients were instructed to spread the oral dose over the whole day in an attempt to decrease the nausea that frequently accompanies the administration of the drug”.

R: We replaced the sentence by the following one:

DAP has been used previously for antileukemia treatment and is reported to exert antiviral^{36, 37, 38, 39} and miRNA inhibition activity⁴⁰.

- Zellshield (page 6) is a mixture of antibiotics. The authors should ascertain whether it contains G418 or aminoglycosides that might affect readthrough.

R: Minerva-Biolabs has provided us with the following composition of Zellshield: macrolides and polyenes, lincosamide and fluoroquinolones.

None of these molecules are aminoglycosides. Although some macrolides have been shown to promote readthrough (Zilberberg et al., 2009 for instance), the molecules present in Zellshield do not have this activity, at least at the concentration used in Zellshield, since when DMSO or no treatment is tested in the presence of Zellshield, no luciferase or readthrough activity is measured or detected (see our Figures 1 and 2 specifically).

- The NMR spectra and other data permitting the identification of the isolated compound as DAP should be provided in appendix. Has the structure of the commercial DAP been ascertained independently?

R: We thank the referee for pointing out the lack of detailed information providing evidence that the molecule of interest isolated from the *L. inversa* extract H7 and 2,6-diamino purine (DAP) are identical.

We have ascertained that natural and commercial synthetic DAP are the same on the following basis:

- a comparison of the HPLC chromatograms obtained for i) the H7 extract, ii) commercial synthetic DAP and iii) the H7 extract co-injected with commercial DAP;

- a comparison of the MS/MS spectra, showing identical fragmentation patterns (characteristic m/z ions) for commercial DAP, the DAP peak selected in the H7 extract, and the isolated natural DAP;

- a comparison of the ¹H and ¹³C NMR data obtained in the ¹H-1D spectrum and the ¹H-¹³C HMBC 2D spectrum, showing the long-range correlations between ¹H and ¹³C signals characteristic of the DAP structure.

Moreover, the structure of commercial DAP (synthetic sample obtained at Sigma-Aldrich; Ref. 247847) is guaranteed by the manufacturer and was nevertheless verified in our laboratory, providing an independent confirmation of the structure.

These data are now provided in Supplementary Materials (Figure S1, S2 and S3 and Table S1).

- It should be stated whether all experiments beyond Fig. 1C were carried out with commercial DAP (as opposed to isolated compound).

R: We have now modified the following sentence:

Given the toxicity issues associated with cliticine⁴⁶, we decided to focus on DAP and, in all subsequent experiments, used commercial DAP only.

- Page 14: to compare the potencies of DAP and G418, their EC50 values should be determined and compared rather than providing statements such as “at 6.25 μ M, it was 10 times as effective as 600 μ M G418”.

R: We have now added the following sentences to clarify this point:

DAP showed maximal efficacy at 100 μ M, with an EC50 of about 50 μ M. For G418, the maximal efficacy was reached at 300 μ M and the EC50 was about 100 μ M. Strikingly, DAP was more than 170 times as effective as G418 at 100 μ M and more than 80 times as effective at 300 μ M.

- Figure 3B showing that DAP increases expression of the Fluc reporter by 2.5-fold over DMSO is weak evidence that p53 produced by readthrough is functional.

R: To demonstrate that the protein synthesized from PTC-containing mRNA under DAP treatment is functional, we have used three assays: (1) testing the p53 protein on firefly luciferase cDNA under the control of a promoter containing p53 response elements; (2) measuring the level of p21 mRNA in Calu-6 cells (and in HeLa cells as a control); (3) measuring the functionality of CFTR in cells collected from cystic fibrosis patients. Besides these three assays, the luciferase activity measured with the Fluc-int-UGA construct further demonstrates that DAP promotes synthesis of a functional readthrough protein (Figure 1).

- I am not qualified to evaluate the mass spectrometry data presented in Fig. 6A but the panel seems difficult to interpret.

R: We have modified the legend of the figure (Fig. 6) in order to make the analysis of the figure easier:

Determination by tandem mass spectrometry of the amino acid incorporated at the PTC position. Firefly luciferase from HEK293FT cells transfected with the Fluc-int-UGA construct was

immunoprecipitated and digested with trypsin. Peptides were analyzed by tandem mass spectrometry. The MS/MS spectrum of the peptide containing the PTC position is shown to determine the sequence. Upper red and blue sequences indicate the amino acids validated, respectively, by detection of Nter and Cter peptide fragments (b and y ion series). Peptide fragments y6 and y7 clearly identify tryptophan (W) at the PTC position.

- In the RiboMethSeq experiment presented in Fig. 6C, it should be indicated what DAP concentration was used and how long the cells were incubated for.

R: We have added this information in the legend of the figure (new 6D):

HeLa cells were incubated for 24 h with DMSO or with 25 μ M DAP.

- Figure 5B showing no DAP modification by S9 liver extract is unclear and the figure is insufficiently labeled. This panel should be moved to supplemental data.

R: We have now moved this panel to Supplemental Figure S7 and have added some labelling to clarify it.

- The last sentence on page 19 (Inhibition of this modification...) should be supported by a reference.

R: The reviewer is right and the sentence was too strong. We have changed it to: "Inhibition of this modification could activate the capacity of tRNA^{Trp} to recognize the UGA stop codon"

Reviewer 2

1. The functional assays are lacking WT controls as a reference. Without comparison to the WT function, the efficiency and level of correction cannot be concluded. This should be performed for the p53 and the CFTR analyses. Only in the mice experiments (Fig 5) a WT p53 control level is shown.

R: We have now included a p53 functional assay with HeLa cells (new figures 3B and 3C). Results are presented in separate panels, as Calu-6 and HeLa cells are not directly comparable because of different genetic backgrounds. For the CFTR assay, our authorization was limited to cystic fibrosis patients carrying nonsense mutations on both alleles encoding CFTR, and did not include the authorization to collect cells from healthy persons.

2. As was shown in the past, each cell type/line has a different NMD efficiency, which affects the level of read-through. The effect of NMD efficiency in each of the studied cellular systems is missing. The authors should include experiments with either NMD inhibitors or siRNA against key NMD factors.

R: The reviewer is right about the influence of NMD on readthrough. We show that DAP does not inhibit NMD (figure 4A). We were careful not to compare the readthrough efficacy of DAP between different cell types. We agree that to do that we would need to normalize the NMD efficiency, probably by inhibiting it, but this will be the focus of another study showing the benefit of using a combination of NMD inhibitors and readthrough activators.

3. The authors conclude that DAP has low toxicity. This is based on two assays. One is the effect of DAP on gene expression. The authors conclude that only 0.13% of the genes showed a change greater than 4 folds. The experiment was performed in HeLa cells. A previous work in HeLa cells by Mendell JT et al., 2004 in Nature Genetics identified ~200 transcripts that are physiologically regulated by NMD. The genes in this group (those that are carrying UGA premature stop) should have been affected by DAP. Moreover, the authors chose a threshold of 4X as a change in expression. This is a too high rough as lower changes may have a profound physiologic effect, as shown in Mendell's paper. I could not find the list of genes that were affected by DAP. It is important to have this list. The authors should present the list of genes that were changes by at least 2 fold. The other assay is measuring AK activity. A more thorough study of cell toxicity should be performed. Assessing cell membrane integrity is one of the most common ways to measure cytotoxic effects. MTT or SRB are other commonly used assays.

R: We have taken into account the comment of reviewer 2 about our statement of low toxicity of DAP towards human cells based on DEGs with a change greater than 4 fold ($|\log_2FC| \geq 2$), which is commonly used to detect some significant biological responses (Li et al., 2018 (DOI: 10.12659/MSM.905410), Liang et al., 2017 (DOI:10.1038/s41598-017-06032-2)). Now, meet the requirement of the reviewer, we have reduced this threshold to 2 fold ($|\log_2FC| \geq 1$) in our new analyses.

The reviewer also compares our results to previously published data of Mendell JT et al, 2004 in Nature Genetics. Mendell found 197 and 176 transcripts which are respectively up- and down-regulated when NMD is inhibited with an siRNA against UPF1. In this study, the authors used microarray technology and analyzed a pool of ~4,000 transcripts (this amounts to ~9 % deregulation). In our study, we used NGS and analyzed the whole genome. NGS analysis is more sensitive than microarray analysis, which explains why we got a similar number of deregulated genes but with a lower biological impact. This is why $|\log_2FC| \geq 2$ is commonly used, rather than a two-fold change in gene expression.

We have taken into account the observation of reviewer 2 concerning the absence of a DEG list. We have added Supplementary Table 1, with the statistics of the 775 DEGs. We have also added a GO term analysis, provided in Supplementary Table 2, and a schematic representation in Supplemental Figure S2C. We have changed the text accordingly:

First, in HeLa cells, we used RNA-seq to assess the impact of DAP on the whole transcription profile. On the basis of statistics alone (adjusted p-value <0.05), 3,228 differentially expressed genes (DEGs) were detected, but no major effect of DAP was observed (Fig. S5). Among the DEGs, 775 showed at least a 2-fold change (adjusted p-value <0.05, $|\log_2FC| \geq 1$ - Table 3)

We also now include the results of two other toxicity assays: a propidium iodide incorporation assay and the MTT assay. Both assays confirm the absence of any significant toxicity of DAP under our experimental conditions (supplemental figure S4).

4. The in vivo experiments in mice presented in Figure 5B are not convincing. The control cells are inadequate, since their tumor development is very different from that in Calu 6 cells with nonsense p53 mutation, as can be seen in the DMSO results. The authors should have chosen cells with the same tumor growth rate. The concern is that DAP may have an anti-proliferative effect and not a specific effect for UGA nonsense mutations.

R: In order to exclude an antiproliferative effect of DAP, we have repeated the experiment with another cell line (4T1 cells), promoting fast tumor growth (Figure 5A) similar to that promoted by Calu-6 cells. We thank the reviewer for pointing out this possibility.

Other comments:

1. Page 17 4 lines from the bottom: what does this mean "no visible differences"? is this an adequate way of analyzing the result?

R: We agree with the comment of the reviewer and have modified the sentence as follows: " The two proteome profiles were found to overlap, without showing any spot shift. This suggests that DAP does not promote synthesis of aberrant proteins (Fig. S6)".

2. Page 18 – the title is misleading since the assay is ex vivo and not in vivo. The ref no 49 is not relevant.

R: This section focuses on mouse experiments, which are in vivo experimentation. The reviewer likely refers to the stability assay on the hepatic extract, which is ex vivo. Since this part of the figure has been moved to "Supporting Information", the title of the section should no longer be confusing.

We have also removed the reference 49 and thank the reviewer for pointing out that inaccurate reference.

3. CFTR functional assay – CFTR function in nasal epithelial cells from patients should have been the Ussing Chamber using differentiated cells. This is the standard functional assay to test the effect of drugs on CFTR function. The assay used by the authors, is the SPQ assay. Where the cells differentiated? No details on the analyses is given and reference is to a review paper.

R: For the SPQ assay, we used cells collected from the nose epithelium of patients. We did not culture them before testing our molecules, in order to test the molecules on the cell model closest to patient cells. Hence, there is no differentiation step before performing the experiment, since the cells are already differentiated. Concerning the reference we used (Mansoura et al., 1999), it is a

technical review describing methods for assessing the function of CFTR, including the SPQ assay. We also used this method and described it in the other reference (Benhabiles et al., 2017).

4. Figure 2 – the first 3 lanes in each of the panels are unclear. What is the identity of the experiment performed in these three lanes?

R: We have added this information in the figure legend: "The three leftmost lanes show twofold serial dilutions of untreated HeLa cell extract."

5. The results presented in Figure 3B and C are inconsistent with the results presented in Figure 2. This includes the G418 different concentrations as well as DAP in comparison to G418. For example, in Figure 3B and C, there is no difference in the effect of G418 using 25 or 600 μ M whereas in the Western blot presented in Figure 2, there is a significant difference in p53 expression between these concentrations. More importantly, 25 μ M DAP has the same effect on the p53 protein level as 600 μ M G418 whereas in Figure 3B and C the effect of these two treatments is very different. These results are confusing. Can the authors explain them?

R: The results in Figure 2 show that both DAP and G418 are able to promote the synthesis of full-length p53 protein, but these data provide no information about the functionality of the full-length p53 protein synthesized. Figures 3B and 3C aim to demonstrate whether the full-length p53 protein is functional or not. Our results show that DAP promotes synthesis of a functional full-length p53 protein; G418, in contrast, promotes synthesis of a non-functional full-length p53 protein.

6. Figure 5C – at what time point was the analyses performed?
What is the 1-7? Different mice? Different days?

R: The tumor size measurements were performed three times per week and the western-blot analysis was performed on tumors collected at the time of sacrifice. We have modified the figure to clarify this and changed the X-axis unit into days after cell injection.

7. Figure 5C shows a very low level of WT p53 in the DAP experiment. This may indicate that the tumor development results are due to a combined effect of DAP on UGA stop and its activity as an anti-proliferative agent.

R: We do not favor the hypothesis of an anti-proliferative effect of DAP on tumor growth, since we observe no such anti-proliferative effect on cells carrying no nonsense mutation (i.e. the HCC-GR6 cells in the previous version of the manuscript and the 4T1 cells in the new version of the manuscript). The low level of p53 detected in tumors is also due to the fact that the TP53 gene is highly regulated at different levels of the gene expression pathway. Although we were able to detect this low-level expression of p53 in mice exposed to DAP, we were unable to detect any expression in mice exposed to DMSO, either after a long exposure period (Figure 5C in the initial version of the manuscript) or after a short period of exposure beginning after the tumor had already developed (new figure 5B)).

8. Figure 5 – the numbers in X axis should be explained. Are these time points of analysis? There were 3 analysis a week for 21 days (according to the text). What are the 17 data points?

R: The reviewer is right, our explanation was not clear and the experiment was not performed for 21 days but for about 5 weeks. We apologize for this and have corrected the text.

9. Proteome analysis – the authors should perform the proteomic analysis on Calu6 and not on HeLa. In order to be able to identify the change in p53.

R: We purposely performed the experiment on cells that do not have nonsense mutations in order to exclude any change in proteomic profile due to re-expression of a mutant gene. By doing the experiment on Calu-6, we would induce expression of the TP53 gene and affect the expression of many other genes. We thus think that the best cell line to use is HeLa and not Calu-6.

10. The manuscript would benefit from English editing

R: the manuscript was originally corrected for English writing and has received a second round of correction before submission of the revised version of the manuscript.

Reviewer 3

First, will DAP cause readthrough of normal UGA stop codons? If it does, it will be toxic and not desirable for the organism. If not, how does it distinguish between UGA nonsense mutation and normal UGA at the end of the ORF?

R: We did not observe production of abnormally long proteins after DAP treatment, as illustrated in the western-blot analysis below and in our 2D gel analysis, since untreated and treated cells displayed identical proteome profiles. To explain why readthrough molecules act on PTCs and not on physiological stop codons, the protein environment is often highlighted: proteins present in the 3'UTR (such as PABPC1) facilitate translation termination at the physiological stop codon, making it very efficient. At a PTC, this protein environment is not present, so translation termination is less efficient and more sensitive to readthrough.

Western-blot analysis as performed in the figure 2 of the manuscript. If DAP promotes the readthrough of physiological UGA stop codon, we should detect some upper bands at less than 1kDa higher than the wild-type p53 length and at 3.5kDa higher than the wild-type p53 size since the ORF of p53 end by a UGA and the next stop codon (at 8 amino-acids after the ORF= about 1kDa) is also a UGA stop codon. Then, a UAG stop codon is present at 24 amino acids after the wild-type ORF (about 3.5kDa). Even with a longer exposure, no upper bands are detected indicating that DAP does not promote readthrough of physiological stop codons.

Another question pertains the effect of DAP on healthy mice. In figure 5B, the effect of DAP on tumor growth in immunodeficient mice was shown. What, if any, would be the effect of DAP on healthy mice?

R: The main goal of Figure 5 is to demonstrate that DAP can promote PTC readthrough in vivo. The question raised by the reviewer is very interesting and deserves emphasis in a study focusing on the therapeutic potential of DAP. It will be integrated into a future study but cannot be part of this manuscript, as the main objective is to report the identification of a new readthrough molecule and its characterization (efficiency, toxicity, and mode of action).

In figure 6C, can you claim significance for 2'-O-methylation of tRNA-Trp in the presence of DAP, since $p=0.066$? Would this cast slight doubt into the effect of DAP in inhibiting methylation of tRNA by FTSJ1?

R: RiboMethSeq analysis of tRNA 2'-O-methylation was performed in biological triplicate and revealed a robust and reproducible decrease of the tRNA^{Trp} modification at position 34 in all 3 independent samples tested (see Figure for individual data points). The p-value ($p=0.066$) calculated with this limited sample number is indeed slightly above the <0.05 , generally accepted in biomedical studies but it is still quite close to the 0.05 limit, and the apparent lack of "statistical significance" is linked to the biological variability of the tRNA^{Trp} Cm34 modification level in untreated cells, not to lack of precision of the RiboMethSeq measurements.

Further, to strengthen this important point of DAP inhibiting FTSJ1 activity, an experiment to determine whether the inhibition is competitive or non-competitive with regard to the methyl donor/tRNA and calculating the K_i value for DAP would be useful.

R: Elucidation of the exact mechanism of FTSJ1 inhibition is certainly of interest, but outside the scope of the present manuscript.

Regarding figure 6B, the text does not describe the western blot data in the right panel. The point of that data is a little unclear to this reviewer.

R: These western-blot analyses are there to show the level of overexpression, and in particular to highlight that the effect observed with FTSJ1 but not with CTU1 was not due to a higher level of FTSJ1 expression as compared to CTU1. We now reference this part of the figure in an independent panel (Figure 6C) and in the following sentence: FTSJ1 and CTU1 overexpression were assessed by western-blot analysis (Fig. 6C). It appeared that overexpression of FTSJ1, but not of CTU1, strongly reduced DAP-promoted readthrough."

Reviewers' comments:

Reviewer #1 (Remarks to the Author):

In this very significantly revised manuscript, the authors have carried out a number of new experiments to address this reviewer's comments and the overall result is much improved.

To address the criticism of the use of an adenylate kinase release assay to measure cell toxicity, the authors have used two additional assays that confirm the lack of significant cytotoxicity even at a high concentration of 2.4 mM, thus addressing an important concern.

To address the criticism that the small levels of p53 detected by western blotting in residual tumour after prolonged DAP treatment could have come from mouse cells, the authors carried out a new in vivo experiment where mice carrying 0.6 cm³ xenografts were treated with DAP for one week prior to determination of p53 by western blotting, using a p53 antibody that does not recognize the mouse protein. They also addressed comment 4 of reviewer 2 by measuring the effect of DAP on a WT p53 tumour xenograft that grows at a similar rate to Calu-6. This section of the paper is now much stronger. Here, the authors have removed the original HCC-GR6 xenograft result and replaced it with the 4T1 result. In my opinion, they should present the HCC-GR6 result in the supplemental section. Similarly, instead of replacing the p53 western blot data originally shown with the new one (Fig. 5C), they should also present the original panel C data in the supplemental section.

The authors have addressed the comment about the small increase in p21 mRNA by 25 μ M DAP (Fig. 3C). Their new comment is speculative but acceptable - the point raised was minor.

The important point that DAP is an adenine analog that can be incorporated into nucleotides has now been discussed adequately.

I commented that in order to compare the potencies of DAP and G418, their EC₅₀ values should be determined and compared rather than providing statements such as "at 6.25 μ M, it was 10 times as effective as 600 μ M G418". This comment was referring to the top panel of Fig. 2. The new statement "DAP showed maximal efficacy at 100 μ M, with an EC₅₀ of about 50 μ M. For G418, the maximal efficacy was reached at 300 μ M and the EC₅₀ was about 100 μ M. Strikingly, DAP was more than 170 times as

effective as G418 at 100 μ M and more than 329 times as effective at 300 μ M” referring to Fig. 1D should be removed in my opinion. It is not possible to compare IC50s in this instance. It would be sufficient to say that G418 is essentially inactive in this assay whereas DAP is highly active.

The manuscript now reads much more clearly and my original very minor comments have also been addressed adequately.

Here are a few additional minor comments:

- Page 3, line 76: “NMD specifically degrades PTC-containing mRNAs” is misleading as most mRNAs degraded by NMD do not contain a PTC. Perhaps replace “specifically” with “can”?
- Page 5, line 111: “when” should be replaced with “whereas”.
- Page 15, line 333 and others: TP53 should be italicized here and elsewhere in the text.
- Page 17, line 393: “adenylated” should be replaced with “adenylate”.
- Page 20, line 462: the sentence is incomplete.
- Page 21, line 479: I believe “confirm that” should be replaced by a more neutral term such as “determine whether” or similar term.
- Page 21, line 480: Fig. 6B should be Fig. 6D.
- Page 23, line 531: the statement that the treated mice did not show “any sign of massive metabolic malfunction” is vague. It should be replaced by more specific wording referring to what was observed in the animal room, such as a comment about the lack of weight loss or signs of ill health (as appropriate)
- Page 25, line 573: “show” could be interpreted as prove or demonstrate. Perhaps “indicate” is a better term.

Reviewer #2 (Remarks to the Author):

Reviewer 2

1. The functional assays are lacking WT controls as a reference. Without comparison to the WT function, the efficiency and level of correction cannot be concluded. This should be performed for the p53 and the CFTR analyses. Only in the mice experiments (Fig 5) a WT p53 control level is shown.

R: We have now included a p53 functional assay with HeLa cells (new figures 3B and 3C). Results are presented in separate panels, as Calu-6 and HeLa cells are not directly comparable because of different genetic backgrounds. For the CFTR assay, our authorization was limited to cystic fibrosis patients carrying nonsense mutations on both alleles encoding CFTR, and did not include the authorization to collect cells from healthy persons.

Reviewer: In order to evaluate the effect magnitude of DAP on readthrough of nonsense mutations, a comparison to WT activity is required.

For p53, the authors now added results from HeLa cells, that do not have a p53 nonsense mutation. The results showed no additional p53 activity following DAP treatment, as expected. However, still there is no way to evaluate the magnitude of DAP effect in Calu-6 cells.

Unfortunately, also the authors did not perform the required additional experiments for CFTR. Thus, evaluation of the effect in Figure 3D is not possible. Furthermore, the results are confusing since patient 1 and 2 differ in their response despite carrying the same genotype (that surprisingly, was omitted from the revised Figure 3D).

2. As was shown in the past, each cell type/line has a different NMD efficiency, which affects the level of read-through. The effect of NMD efficiency in each of the studied cellular systems is missing.

The authors should include experiments with either NMD inhibitors or siRNA against key NMD factors.

R: The reviewer is right about the influence of NMD on readthrough. We show that DAP does not inhibit NMD (figure 4A). We were careful not to compare the readthrough efficacy of DAP between different cell types. We agree that to do that we would need to normalize the NMD efficiency, probably by

inhibiting it, but this will be the focus of another study showing the benefit of using a combination of NMD inhibitors and readthrough activators.

Reviewer: Unfortunately, the authors did not perform the required experiments.

3. The authors conclude that DAP has low toxicity. This is based on two assays. One is the effect of DAP on gene expression. The authors conclude that only 0.13% of the genes showed a change greater than 4 folds. The experiment was performed in HeLa cells. A previous work in HeLa cells by Mendell JT et al., 2004 in Nature Genetics identified ~200 transcripts that are physiologically regulated by NMD. The genes in this group (those that are carrying UGA premature stop) should have been affected by DAP. Moreover, the authors chose a threshold of 4X as a change in expression. This is a too high rough as lower changes may have a profound physiologic effect, as shown in Mendell's paper. I could not find the list of genes that were affected by DAP. It is important to have this list. The authors should present the list of genes that were changes by at least 2 fold.

The other assay is measuring AK activity. A more thorough study of cell toxicity should be performed.

Assessing cell membrane integrity is one of the most common ways to measure cytotoxic effects.

MTT or SRB are other commonly used assays.

R: We have taken into account the comment of reviewer 2 about our statement of low toxicity of DAP towards human cells based on DEGs with a change greater than 4 fold ($|\log_2FC| \geq 2$), which is commonly used to detect some significant biological responses (Li et al., 2018 (DOI: 10.12659/MSM.905410), Liang et al., 2017 (DOI:10.1038/s41598-017-06032-2)). Now, meet the requirement of the reviewer, we have reduced this threshold to 2 fold ($|\log_2FC| \geq 1$) in our new analyses.

OK

R: The reviewer also compares our results to previously published data of Mendell JT et al, 2004 in Nature Genetics. Mendell found 197 and 176 transcripts which are respectively up- and downregulated when NMD is inhibited with an siRNA against UPF1. In this study, the authors used microarray technology and analyzed a pool of ~4,000 transcripts (this amounts to ~9 %

deregulation). In our study, we used NGS and analyzed the whole genome. NGS analysis is more sensitive than microarray analysis, which explains why we got a similar number of deregulated genes but with a lower biological impact. This is why $|\log_2FC| \geq 2$ is commonly used, rather than a two-fold change in gene expression.

We have taken into account the observation of reviewer 2 concerning the absence of a DEG list. We have added Supplementary Table 1, with the statistics of the 775 DEGs. We have also added a GO term analysis, provided in Supplementary Table 2, and a schematic representation in Supplemental Figure S2C. We have changed the text accordingly:

First, in HeLa cells, we used RNA-seq to assess the impact of DAP on the whole transcription profile. On the basis of statistics alone (adjusted p-value < 0.05), 3,228 differentially expressed genes (DEGs) were detected, but no major effect of DAP was observed (Fig. S5).

Among the DEGs, 775 showed at least a 2-fold change (adjusted p-value < 0.05 , $|\log_2FC| \geq 1$

- Table 3)

Reviewer: The authors included the list of genes that were changed by DAP. They have also reduced the threshold.

However, the conclusion of the authors: " Altogether, these results indicate unambiguously that DAP does not interfere with the expression program of HeLa cells at either the mRNA or protein level." is over looking on the effect of DAP on other genes, as expression changes of hundreds of genes is not marginal and raises a significant concern.

R: We also now include the results of two other toxicity assays: a propidium iodide incorporation assay and the MTT assay. Both assays confirm the absence of any significant toxicity of DAP under our experimental conditions (supplemental figure S4).

Reviewer: The authors now provided the required information.

4. The in vivo experiments in mice presented in Figure 5B are not convincing. The control cells are inadequate, since their tumor development is very different from that in Calu 6 cells with nonsense p53 mutation, as can be seen in the DMSO results. The authors should have chosen cells with the same tumor growth rate. The concern is that DAP may have an anti-proliferative effect and not a specific effect for UGA nonsense mutations.

R: In order to exclude an antiproliferative effect of DAP, we have repeated the experiment with another cell line (4T1 cells), promoting fast tumor growth (Figure 5A) similar to that promoted by Calu-6 cells. We thank the reviewer for pointing out this possibility.

Reviewer: The addition of 4T1 cells is a good choice.

However, the Western results presented in the original Figure 5C was replaced with another Western, in which the results in lane 7 of DAP, now show a very low effect. What is the significant of these results?

Other comments:

1. Page 17 4 lines from the bottom: what does this mean "no visible differences"? is this an adequate way of analyzing the result?

R: We agree with the comment of the reviewer and have modified the sentence as follows: " The two proteome profiles were found to overlap, without showing any spot shift. This suggests that DAP does not promote synthesis of aberrant proteins (Fig. S6)".

Reviewer: The overlap analysis is insufficiently sensitive to conclude that there are no differences.

2. Page 18 – the title is misleading since the assay is ex vivo and not in vivo. The ref no 49 is not relevant.

R: This section focuses on mouse experiments, which are in vivo experimentation. The reviewer likely refers to the stability assay on the hepatic extract, which is ex vivo. Since this part of the figure has been moved to “Supporting Information”, the title of the section should no longer be confusing.

We have also removed the reference 49 and thank the reviewer for pointing out that inaccurate reference.

Reviewer: OK

3. CFTR functional assay – CFTR function in nasal epithelial cells from patients should have been the Ussing Chamber using differentiated cells. This is the standard functional assay to test the effect of drugs on CFTR function. The assay used by the authors, is the SPQ assay. Where the cells differentiated? No details on the analyses is given and reference is to a review paper.

R: For the SPQ assay, we used cells collected from the nose epithelium of patients. We did not culture them before testing our molecules, in order to test the molecules on the cell model closest to patient cells. Hence, there is no differentiation step before performing the experiment, since the cells are already differentiated. Concerning the reference we used (Mansoura et al., 1999), it is a technical review describing methods for assessing the function of CFTR, including the SPQ assay. We also used this method and described it in the other reference (Benhabiles et al., 2017).

Reviewer: The authors did not perform the required experiment. As explained in the previous comment the authors should have used the gold standard assay, the Ussing chamber measurements for studying the effect of DAP on nonsense mutations in the CFTR gene. This is important in order to compare the effect of DAP to the effect of other tested molecules.

4. Figure 2 – the first 3 lanes in each of the panels are unclear. What is the identity of the experiment performed in these three lanes?

R: We have added this information in the figure legend: " The three leftmost lanes show twofold serial dilutions of untreated HeLa cell extract."

Reviewer: The authors now explain what is presented in lanes 1-3. Hence, the results from the Western indicate that DAP has a significant effect when 100-600 μ M are used, comparable to the lowest level in HeLa cells. Thus, the correction in Figure 3 using 6.25 or 25 μ M, have a low effect on the protein level. This Western should be included in Figure 3, and the results should be discussed accordingly.

5. The results presented in Figure 3B and C are inconsistent with the results presented in Figure 2. This includes the G418 different concentrations as well as DAP in comparison to G418. For example, in Figure 3B and C, there is no difference in the effect of G418 using 25 or 600 μ M whereas in the Western blot presented in Figure 2, there is a significant difference in p53 expression between these concentrations. More importantly, 25 μ M DAP has the same effect on the p53 protein level as 600 μ M G418 whereas in Figure 3B and C the effect of these two treatments is very different. These results are confusing. Can the authors explain them?

R: The results in Figure 2 show that both DAP and G418 are able to promote the synthesis of full-length p53 protein, but these data provide no information about the functionality of the full-length p53 protein synthesized. Figures 3B and 3C aim to demonstrate whether the full-length p53 protein is functional or not. Our results show that DAP promotes synthesis of a functional full-length p53 protein; G418, in contrast, promotes synthesis of a non-functional full-length p53 protein.

Reviewer: The authors do not explain the inconsistent results detailed in this comment. Moreover, it is puzzling that G418 readthrough results in a non-functional proteins whereas DAP results in a functional protein, as the two reagents presents the same protein level

6. Figure 5C – at what time point was the analyses performed?

What is the 1-7? Different mice? Different days?

R: The tumor size measurements were performed three times per week and the western-blot analysis was performed on tumors collected at the time of sacrifice. We have modified the figure to clarify this and changed the X-axis unit into days after cell injection.

Reviewer: OK

7. Figure 5C shows a very low level of WT p53 in the DAP experiment. This may indicate that the tumor development results are due to a combined effect of DAP on UGA stop and its activity as an anti-proliferative agent.

R: We do not favor the hypothesis of an anti-proliferative effect of DAP on tumor growth, since we observe no such anti-proliferative effect on cells carrying no nonsense mutation (i.e. the HCC-GR6 cells in the previous version of the manuscript and the 4T1 cells in the new version of the manuscript). The low level of p53 detected in tumors is also due to the fact that the TP53 gene is highly regulated at different levels of the gene expression pathway. Although we were able to detect this low-level expression of p53 in mice exposed to DAP, we were unable to detect any expression in mice exposed to DMSO, either after a long exposure period (Figure 5C in the initial version of the manuscript) or after a short period of exposure beginning after the tumor had already developed (new figure 5B)).

Reviewer: The revised figure 5 is lacking a control. The authors should include as a control 4T1.

8. Figure 5 – the numbers in X axis should be explained. Are these time points of analysis? There were 3 analysis a week for 21 days (according to the text). What are the 17 data points?

R: The reviewer is right, our explanation was not clear and the experiment was not performed for 21 days but for about 5 weeks. We apologize for this and have corrected the text.

Reviewer: OK

9. Proteome analysis – the authors should perform the proteomic analysis on Calu6 and not on HeLa.

In order to be able to identify the change in p53.

R: We purposely performed the experiment on cells that do not have nonsense mutations in order to exclude any change in proteomic profile due to re-expression of a mutant gene. By doing the experiment on Calu-6, we would induce expression of the TP53 gene and affect the expression of many other genes. We thus think that the best cell line to use is HeLa and not Calu-6.

Reviewer: As all the experiments were performed in Calu-6, the proteome analysis should have been performed in these cells.

10. The manuscript would benefit from English editing

R: the manuscript was originally corrected for English writing and has received a second round of correction before submission of the revised version of the manuscript.

Reviewer: OK

Reviewer #3 (Remarks to the Author):

I am happy with the revised version. I hoped that they would go into preliminary studies on the mechanism of DAP inhibition of the methylase. Namely, how DAP inhibits the methylation. However, that subject would probably be the subject of the next publication. I am satisfied with the current version of the manuscript.

We would like to thank the three reviewers for their time and their comments on the revised version of our manuscript. Here are our answers that we hope will address their last concerns.

Reviewers' comments:

Reviewer #1 (Remarks to the Author):

In this very significantly revised manuscript, the authors have carried out a number of new experiments to address this reviewer's comments and the overall result is much improved.

To address the criticism of the use of an adenylate kinase release assay to measure cell toxicity, the authors have used two additional assays that confirm the lack of significant cytotoxicity even at a high concentration of 2.4 mM, thus addressing an important concern.

To address the criticism that the small levels of p53 detected by western blotting in residual tumour after prolonged DAP treatment could have come from mouse cells, the authors carried out a new in vivo experiment where mice carrying 0.6 cm³ xenografts were treated with DAP for one week prior to determination of p53 by western blotting, using a p53 antibody that does not recognize the mouse protein. They also addressed comment 4 of reviewer 2 by measuring the effect of DAP on a WT p53 tumour xenograft that grows at a similar rate to Calu-6. This section of the paper is now much stronger. Here, the authors have removed the original HCC-GR6 xenograft result and replaced it with the 4T1 result. In my opinion, they should present the HCC-GR6 result in the supplemental section. Similarly, instead of replacing the p53 western blot data originally shown with the new one (Fig. 5C), they should also present the original panel C data in the supplemental section.

R: We added the original figure as supplemental figure S8.

The authors have addressed the comment about the small increase in p21 mRNA by 25 μ M DAP (Fig. 3C). Their new comment is speculative but acceptable - the point raised was minor.

The important point that DAP is an adenine analog that can be incorporated into nucleotides has now been discussed adequately.

I commented that in order to compare the potencies of DAP and G418, their EC₅₀ values should be determined and compared rather than providing statements such as "at 6.25 μ M, it was 10 times as effective as 600 μ M G418". This comment was referring to the top panel of Fig. 2. The new statement "DAP showed maximal efficacy at 100 μ M, with an EC₅₀ of about 50 μ M. For G418, the maximal efficacy was reached at 300 μ M and the EC₅₀ was about 100 μ M. Strikingly, DAP was more than 170 times as effective as G418 at 100 μ M and more than 329 times as effective at 300 μ M" referring to Fig. 1D should be removed in my opinion. It is not possible to compare IC₅₀s in this instance. It would be sufficient to say that G418 is essentially inactive in this assay whereas DAP is highly active.

R: We modified the text as following: "Increasing amounts of G418 in the cell culture medium promoted a modest increase of luciferase activity (about two times the background level) unlike increasing amounts of DAP that demonstrated a higher capacity in UGA readthrough efficiency." In addition, we added a panel in the figure 1D with the luciferase activity values for each point in order to help the reader to evaluate the capacity of G418 and DAP to readthrough UGA PTC.

The manuscript now reads much more clearly and my original very minor comments have also been addressed adequately.

Here are a few additional minor comments:

- Page 3, line 76: "NMD specifically degrades PTC-containing mRNAs" is misleading as most mRNAs degraded by NMD do not contain a PTC. Perhaps replace "specifically" with "can"?
- Page 5, line 111: "when" should be replaced with "whereas".
- Page 15, line 333 and others: TP53 should be italicized here and elsewhere in the text.
- Page 17, line 393: "adenylated" should be replaced with "adenylate".
- Page 20, line 462: the sentence is incomplete.
- Page 21, line 479: I believe "confirm that" should be replaced by a more neutral term such as "determine whether" or similar term.
- Page 21, line 480: Fig. 6B should be Fig. 6D.
- Page 23, line 531: the statement that the treated mice did not show "any sign of massive metabolic malfunction" is vague. It should be replaced by more specific wording referring to what was observed in the animal room, such as a comment about the lack of weight loss or signs of ill health (as appropriate)
- Page 25, line 573: "show" could be interpreted as prove or demonstrate. Perhaps "indicate" is a better term.

R: we replaced as requested and thank this reviewer for the corrections.

Reviewer #2 (Remarks to the Author):

Reviewer 2

1. The functional assays are lacking WT controls as a reference. Without comparison to the WT function, the efficiency and level of correction cannot be concluded. This should be performed for the p53 and the CFTR analyses. Only in the mice experiments (Fig 5) a WT p53 control level is shown.

R: We have now included a p53 functional assay with HeLa cells (new figures 3B and 3C). Results are presented in separate panels, as Calu-6 and HeLa cells are not directly comparable because of different genetic backgrounds. For the CFTR assay, our authorization was limited to cystic fibrosis

patients carrying nonsense mutations on both alleles encoding CFTR, and did not include the authorization to collect cells from healthy persons.

Reviewer: In order to evaluate the effect magnitude of DAP on readthrough of nonsense mutations, a comparison to WT activity is required.

For p53, the authors now added results from HeLa cells, that do not have a p53 nonsense mutation. The results showed no additional p53 activity following DAP treatment, as expected. However, still there is no way to evaluate the magnitude of DAP effect in Calu-6 cells.

Unfortunately, also the authors did not perform the required additional experiments for CFTR. Thus, evaluation of the effect in Figure 3D is not possible. Furthermore, the results are confusing since patient 1 and 2 differ in their response despite carrying the same genotype (that surprisingly, was omitted from the revised Figure 3D).

R: We totally understand the issue raised by the reviewer. We removed the results concerning the rescue of the function of CFTR in patient cells since our authorization ended and did not cover the collect of cells from healthy people.

Concerning the cancer cell lines we added the following sentence to the discussion since the concern raised by the reviewer is impossible to address due to the lack of isogenic wild-type cells and the use of transfected construct would be inaccurate due to variability of gene expression:

“Although DAP appears to be more potent than G418, we were not able to calculate the rescue of the gene expression compared to wild-type since no isogenic wild-type cells corresponding to our used cell lines exist.”

2. As was shown in the past, each cell type/line has a different NMD efficiency, which affects the level of read-through. The effect of NMD efficiency in each of the studied cellular systems is missing. The authors should include experiments with either NMD inhibitors or siRNA against key NMD factors.

R: The reviewer is right about the influence of NMD on readthrough. We show that DAP does not inhibit NMD (figure 4A). We were careful not to compare the readthrough efficacy of DAP between different cell types. We agree that to do that we would need to normalize the NMD efficiency, probably by inhibiting it, but this will be the focus of another study showing the benefit of using a combination of NMD inhibitors and readthrough activators.

Reviewer: Unfortunately, the authors did not perform the required experiments.

R: The experiments suggested by the reviewer are very interesting but are out of the scope of this study that focuses on the characterization of DAP as a readthrough activator. Our study does not focus on the optimization of the efficiency of DAP that is already very high. The reviewer experiments would require to identify an NMD inhibitor compatible with DAP treatment and we know that only few NMD inhibitors have been identified since we are actively involved in this domain too. Then, the combination DAP with NMD inhibitor would have to be used to test on cell

models and in vivo in mouse model in order to get the information about whether or not it is possible to obtain a higher rescue efficiency on the expression of PTC-containing genes. This information is not necessary for this first study on DAP that has to show its capacity at correcting UGA nonsense mutation by itself. The combination will be included in a future study focusing on the therapeutic aspects of DAP and the way to optimize the correction of nonsense mutation by DAP.

3. The authors conclude that DAP has low toxicity. This is based on two assays. One is the effect of DAP on gene expression. The authors conclude that only 0.13% of the genes showed a change greater than 4 folds. The experiment was performed in HeLa cells. A previous work in HeLa cells by Mendell JT et al., 2004 in Nature Genetics identified ~200 transcripts that are physiologically regulated by NMD. The genes in this group (those that are carrying UGA premature stop) should have been affected by DAP. Moreover, the authors chose a threshold of 4X as a change in expression. This is a too high rough as lower changes may have a profound physiologic effect, as shown in Mendell's paper. I could not find the list of genes that were affected by DAP. It is important to have this list. The authors should present the list of genes that were changes by at least 2 fold. The other assay is measuring AK activity. A more thorough study of cell toxicity should be performed. Assessing cell membrane integrity is one of the most common ways to measure cytotoxic effects. MTT or SRB are other commonly used assays.

R: We have taken into account the comment of reviewer 2 about our statement of low toxicity of DAP towards human cells based on DEGs with a change greater than 4 fold ($|\log_2FC| \geq 2$), which is commonly used to detect some significant biological responses (Li et al., 2018 (DOI: 10.12659/MSM.905410), Liang et al., 2017 (DOI:10.1038/s41598-017-06032-2)). Now, meet the requirement of the reviewer, we have reduced this threshold to 2 fold ($|\log_2FC| \geq 1$) in our new analyses.

OK

R: The reviewer also compares our results to previously published data of Mendell JT et al, 2004 in Nature Genetics. Mendell found 197 and 176 transcripts which are respectively up- and downregulated when NMD is inhibited with an siRNA against UPF1. In this study, the authors used microarray technology and analyzed a pool of ~4,000 transcripts (this amounts to ~9 % deregulation). In our study, we used NGS and analyzed the whole genome. NGS analysis is more sensitive than microarray analysis, which explains why we got a similar number of deregulated genes but with a lower biological impact. This is why $|\log_2FC| \geq 2$ is commonly used, rather than a two-fold change in gene expression.

We have taken into account the observation of reviewer 2 concerning the absence of a DEG list. We have added Supplementary Table 1, with the statistics of the 775 DEGs. We have also added a GO term analysis, provided in Supplementary Table 2, and a schematic representation in Supplemental Figure S2C. We have changed the text accordingly:

First, in HeLa cells, we used RNA-seq to assess the impact of DAP on the whole transcription profile. On the basis of statistics alone (adjusted p-value < 0.05), 3,228 differentially expressed genes (DEGs) were detected, but no major effect of DAP was observed (Fig. S5). Among the DEGs, 775 showed at least a 2-fold change (adjusted p-value < 0.05 , $|\log_2FC| \geq 1$

- Table 3)

Reviewer: The authors included the list of genes that were changed by DAP. They have also reduced the threshold.

However, the conclusion of the authors: " Altogether, these results indicate unambiguously that DAP does not interfere with the expression program of HeLa cells at either the mRNA or protein level." is over looking on the effect of DAP on other genes, as expression changes of hundreds of genes is not marginal and raises a significant concern.

R: We modified the text as following: "Altogether, these results indicate that DAP has a moderate impact on the gene expression program and affects mainly pathways to produce energy that could reflect the need for the activation of readthrough."

R: We also now include the results of two other toxicity assays: a propidium iodide incorporation assay and the MTT assay. Both assays confirm the absence of any significant toxicity of DAP under our experimental conditions (supplemental figure S4).

Reviewer: The authors now provided the required information.

4. The in vivo experiments in mice presented in Figure 5B are not convincing. The control cells are inadequate, since their tumor development is very different from that in Calu 6 cells with nonsense p53 mutation, as can be seen in the DMSO results. The authors should have chosen cells with the same tumor growth rate. The concern is that DAP may have an anti-proliferative effect and not a specific effect for UGA nonsense mutations.

R: In order to exclude an antiproliferative effect of DAP, we have repeated the experiment with another cell line (4T1 cells), promoting fast tumor growth (Figure 5A) similar to that promoted by Calu-6 cells. We thank the reviewer for pointing out this possibility.

Reviewer: The addition of 4T1 cells is a good choice.

However, the Western results presented in the original Figure 5C was replaced with another Western, in which the results in lane 7 of DAP, now show a very low effect. What is the significant of these results?

R: I am not sure to understand the comment of the reviewer since on figure 5B, the band intensity in the lane 7 for DAP treatment is quite similar to the band intensity in the lane 6 for instance. Since the experiment is new made from new mice samples, we cannot compare the intensity bands with two different experiments. In addition, I would like to remind the reviewer that mice were exposed during a short period of time in the new experiment. Therefore, the rescue of p53 level might be lower than the rescue level observed after several weeks of exposure.

Other comments:

1. Page 17 4 lines from the bottom: what does this mean "no visible differences"? is this an adequate

way of analyzing the result?

R: We agree with the comment of the reviewer and have modified the sentence as follows: " The two proteome profiles were found to overlap, without showing any spot shift. This suggests that DAP does not promote synthesis of aberrant proteins (Fig. S6)".

Reviewer: The overlap analysis is insufficiently sensitive to conclude that there are no differences.

R: We now provide a Pearson coefficient which is 0.94. The correlation value ranges between 1 and -1. A value of 1 represents perfect correlation, 0 no correlation, and -1 perfect inverse correlation. In addition, this experiment is a 2D-DIGE experiment which is more sensitive than a 2D gel followed by a coomassie staining provided in other studies (Welch et al., 2007; DOI nature05756 [pii]10.1038/nature05756).

2. Page 18 – the title is misleading since the assay is ex vivo and not in vivo. The ref no 49 is not relevant.

R: This section focuses on mouse experiments, which are in vivo experimentation. The reviewer likely refers to the stability assay on the hepatic extract, which is ex vivo. Since this part of the figure has been moved to "Supporting Information", the title of the section should no longer be confusing.

We have also removed the reference 49 and thank the reviewer for pointing out that inaccurate reference.

Reviewer: OK

3. CFTR functional assay – CFTR function in nasal epithelial cells from patients should have been the Ussing Chamber using differentiated cells. This is the standard functional assay to test the effect of drugs on CFTR function. The assay used by the authors, is the SPQ assay. Where the cells differentiated? No details on the analyses is given and reference is to a review paper.

R: For the SPQ assay, we used cells collected from the nose epithelium of patients. We did not culture them before testing our molecules, in order to test the molecules on the cell model closest to patient cells. Hence, there is no differentiation step before performing the experiment, since the cells are already differentiated. Concerning the reference we used (Mansoura et al., 1999), it is a technical review describing methods for assessing the function of CFTR, including the SPQ assay. We also used this method and described it in the other reference (Benhabiles et al., 2017).

Reviewer: The authors did not perform the required experiment. As explained in the previous comment the authors should have used the gold standard assay, the Ussing chamber measurements for studying the effect of DAP on nonsense mutations in the CFTR gene. This is important in order to compare the effect of DAP to the effect of other tested molecules.

R: To solve this issue that can not be experimentally solved since we do not have anymore these rare patient cells, we removed this experiment.

4. Figure 2 – the first 3 lanes in each of the panels are unclear. What is the identity of the experiment performed in these three lanes?

R: We have added this information in the figure legend: " The three leftmost lanes show twofold serial dilutions of untreated HeLa cell extract."

Reviewer: The authors now explain what is presented in lanes 1-3. Hence, the results from the Western indicate that DAP has a significant effect when 100-600 μM are used, comparable to the lowest level in HeLa cells. Thus, the correction in Figure 3 using 6.25 or 25 μM , have a low effect on the protein level. This Western should be included in Figure 3, and the results should be discussed accordingly.

R: We now have merged the results of the figure 2 and 3 in order to generate the new figure 2. In addition, we added in the discussion the following sentences to indicate why we used 25 μM of DAP as treatment and why the functional rescue might be suboptimal: " In addition, 25 μM was used as the highest concentration of DAP in most of our experiments to characterize the readthrough activity of DAP in order to reduce the risk of an eventual DAP toxicity. As a consequence, the rescue of the function of genes carrying a UGA nonsense mutation by DAP shown in our study (Fig. 2) is likely suboptimal. "

5. The results presented in Figure 3B and C are inconsistent with the results presented in Figure 2. This includes the G418 different concentrations as well as DAP in comparison to G418. For example, in Figure 3B and C, there is no difference in the effect of G418 using 25 or 600 μM whereas in the Western blot presented in Figure 2, there is a significant difference in p53 expression between these concentrations. More importantly, 25 μM DAP has the same effect on the p53 protein level as 600 μM G418 whereas in Figure 3B and C the effect of these two treatments is very different. These results are confusing. Can the authors explain them?

R: The results in Figure 2 show that both DAP and G418 are able to promote the synthesis of fulllength p53 protein, but these data provide no information about the functionality of the full-length p53 protein synthesized. Figures 3B and 3C aim to demonstrate whether the full-length p53 protein is functional or not. Our results show that DAP promotes synthesis of a functional full-length p53 protein; G418, in contrast, promotes synthesis of a non-functional full-length p53 protein.

Reviewer: The authors do not explain the inconsistent results detailed in this comment. Moreover, it is puzzling that G418 readthrough results in a non-functional proteins whereas DAP results in a functional protein, as the two reagents presents the same protein level

R: I apologize for not being clear. My point is that the synthesis of a full length protein is not a warranty of producing a functional protein because the amino-acid introduce at the PTC position

might not be compatible with the function of the protein. G418 and DAP have different mode of action and when DAP will promote the insertion of a tryptophan, G418 might induce the introduction of another amino acid at that position making the full-size protein a not functional protein. It is not a question of amount of protein but more about the composition in amino acid that will make the difference in figure 2 panels C, D and E.

6. Figure 5C – at what time point was the analyses performed?
What is the 1-7? Different mice? Different days?

R: The tumor size measurements were performed three times per week and the western-blot analysis was performed on tumors collected at the time of sacrifice. We have modified the figure to clarify this and changed the X-axis unit into days after cell injection.

Reviewer: OK

7. Figure 5C shows a very low level of WT p53 in the DAP experiment. This may indicate that the tumor development results are due to a combined effect of DAP on UGA stop and its activity as an anti-proliferative agent.

R: We do not favor the hypothesis of an anti-proliferative effect of DAP on tumor growth, since we observe no such anti-proliferative effect on cells carrying no nonsense mutation (i.e. the HCC-GR6 cells in the previous version of the manuscript and the 4T1 cells in the new version of the manuscript). The low level of p53 detected in tumors is also due to the fact that the TP53 gene is highly regulated at different levels of the gene expression pathway. Although we were able to detect this low-level expression of p53 in mice exposed to DAP, we were unable to detect any expression in mice exposed to DMSO, either after a long exposure period (Figure 5C in the initial version of the manuscript) or after a short period of exposure beginning after the tumor had already developed (new figure 5B)).

Reviewer: The revised figure 5 is lacking a control. The authors should include as a control 4T1.

R: the reason why 4T1 cells were not included in the p53 western-blot is that the antibody used to detect p53 recognizes only human p53 and not mouse p53 in order to prevent the detection of any mouse tissue contamination in the tumor analysis as requested. In addition, 4T1 cells do not express p53 protein.

8. Figure 5 – the numbers in X axis should be explained. Are these time points of analysis? There were 3 analysis a week for 21 days (according to the text). What are the 17 data points?

R: The reviewer is right, our explanation was not clear and the experiment was not performed for 21 days but for about 5 weeks. We apologize for this and have corrected the text.

Reviewer: OK

9. Proteome analysis – the authors should perform the proteomic analysis on Calu6 and not on HeLa.

In order to be able to identify the change in p53.

R: We purposely performed the experiment on cells that do not have nonsense mutations in order to exclude any change in proteomic profile due to re-expression of a mutant gene. By doing the experiment on Calu-6, we would induce expression of the TP53 gene and affect the expression of many other genes. We thus think that the best cell line to use is HeLa and not Calu-6.

Reviewer: As all the experiments were performed in Calu-6, the proteome analysis should have been performed in these cells.

R: This experiment was performed in order to determine whether DAP interfere with the general protein expression profile. Making the proteome analysis in Calu-6 would make the analysis and the conclusion inaccurate because of the rescue of TP53 expression would impact the expression of many genes. We would expect to have a modification of the proteome profile due to this p53 expression rescue and not due to the general readthrough activity of DAP which is the aim of this experiment. We therefore chose to perform it in HeLa cells as we did to determine the impact of DAP on the general transcription profile (Fig. S5).

10. The manuscript would benefit from English editing

R: the manuscript was originally corrected for English writing and has received a second round of correction before submission of the revised version of the manuscript.

Reviewer: OK

Reviewer #3 (Remarks to the Author):

I am happy with the revised version. I hoped that they would go into preliminary studies on the mechanism of DAP inhibition of the methylase. Namely, how DAP inhibits the methylation. However, that subject would probably be the subject of the next publication. I am satisfied with the current version of the manuscript.

R: We thank the reviewer for raising interesting questions and insure him/her that we will go on with the study of DAP.

REVIEWERS' COMMENTS:

Reviewer #2 (Remarks to the Author):

We would like to thank the three reviewers for their time and their comments on the revised version of our manuscript. Here are our answers that we hope will address their last concerns.

Reviewers' comments:

Reviewer 2

1. The functional assays are lacking WT controls as a reference. Without comparison to the WT function, the efficiency and level of correction cannot be concluded. This should be performed for the p53 and the CFTR analyses. Only in the mice experiments (Fig 5) a WT p53 control level is shown.

R: We have now included a p53 functional assay with HeLa cells (new figures 3B and 3C). Results are presented in separate panels, as Calu-6 and HeLa cells are not directly comparable because of different genetic backgrounds. For the CFTR assay, our authorization was limited to cystic fibrosis patients carrying nonsense mutations on both alleles encoding CFTR, and did not include the authorization to collect cells from healthy persons.

Reviewer: In order to evaluate the effect magnitude of DAP on readthrough of nonsense mutations, a comparison to WT activity is required.

For p53, the authors now added results from HeLa cells, that do not have a p53 nonsense mutation. The results showed no additional p53 activity following DAP treatment, as expected. However, still there is no way to evaluate the magnitude of DAP effect in Calu-6 cells.

Unfortunately, also the authors did not perform the required additional experiments for CFTR. Thus, evaluation of the effect in Figure 3D is not possible. Furthermore, the results are confusing since patient 1 and 2 differ in their response despite carrying the same genotype (that surprisingly, was

omitted from the revised Figure 3D).

R: We totally understand the issue raised by the reviewer. We removed the results concerning the rescue of the function of CFTR in patient cells since our authorization ended and did not cover the collect of cells from healthy people.

Reviewer: It is unfortunate that no control experiments could be performed in CFTR mutated cells. Having said that, without appropriate controls the results can not be evaluated.

Concerning the cancer cell lines we added the following sentence to the discussion since the concern raised by the reviewer is impossible to address due to the lack of isogenic wild-type cells and the use of transfected construct would be inaccurate due to variability of gene expression:

“Although DAP appears to be more potent than G418, we were not able to calculate the rescue of the gene expression compared to wild-type since no isogenic wild-type cells corresponding to our used cell lines exist.”

Reviewer: Thus, the effect of DAP in cells carrying a p53 nonsense mutation cannot be evaluated.

2. As was shown in the past, each cell type/line has a different NMD efficiency, which affects the level of read-through. The effect of NMD efficiency in each of the studied cellular systems is missing. The authors should include experiments with either NMD inhibitors or siRNA against key NMD factors.

R: The reviewer is right about the influence of NMD on readthrough. We show that DAP does not inhibit NMD (figure 4A). We were careful not to compare the readthrough efficacy of DAP between different cell types. We agree that to do that we would need to normalize the NMD efficiency, probably by inhibiting it, but this will be the focus of another study showing the benefit of using a combination of NMD inhibitors and readthrough activators.

Reviewer: Unfortunately, the authors did not perform the required experiments.

R: The experiments suggested by the reviewer are very interesting but are out of the scope of this study that focuses on the characterization of DAP as a readthrough activator. Our study does not focus on the optimization of the efficiency of DAP that is already very high. The reviewer experiments would require to identify an NMD inhibitor compatible with DAP treatment and we know that only few NMD inhibitors have been identified since we are actively involved in this domain too. Then, the combination DAP with NMD inhibitor would have to be used to test on cell models and in vivo in mouse model in order to get the information about whether or not it is possible to obtain a higher rescue efficiency on the expression of PTC-containing genes. This information is not necessary for this first study on DAP that has to show its capacity at correcting UGA nonsense mutation by itself. The combination will be included in a future study focusing on the therapeutic aspects of DAP and the way to optimize the correction of nonsense mutation by DAP.

Reviewer: It is not clear why the authors claims that NMD inhibitor should be compatible with DAP. Several NMD inhibitors are available and widely used among them: SMG1 inhibitor, NMD1, siRNA against key NMD factors among others. The NMD experiments are expected to significantly add to the manuscript by revealing the total effect of DAP.

3. The authors conclude that DAP has low toxicity. This is based on two assays. One is the effect of DAP on gene expression. The authors conclude that only 0.13% of the genes showed a change greater than 4 folds. The experiment was performed in HeLa cells. A previous work in HeLa cells by Mendell JT et al., 2004 in Nature Genetics identified ~200 transcripts that are physiologically regulated by NMD. The genes in this group (those that are carrying UGA premature stop) should have been affected by DAP. Moreover, the authors chose a threshold of 4X as a change in expression. This is a too high rough as lower changes may have a profound physiologic effect, as shown in Mendell's paper. I could not find the list of genes that were affected by DAP. It is important to have this list. The

authors should present the list of genes that were changes by at least 2 fold.

The other assay is measuring AK activity. A more thorough study of cell toxicity should be performed.

Assessing cell membrane integrity is one of the most common ways to measure cytotoxic effects.

MTT or SRB are other commonly used assays.

R: We have taken into account the comment of reviewer 2 about our statement of low toxicity of DAP towards human cells based on DEGs with a change greater than 4 fold ($|\log_2FC| \geq 2$), which is commonly used to detect some significant biological responses (Li et al., 2018 (DOI: 10.12659/MSM.905410), Liang et al., 2017 (DOI:10.1038/s41598-017-06032-2)). Now, meet the requirement of the reviewer, we have reduced this threshold to 2 fold ($|\log_2FC| \geq 1$) in our new analyses.

OK

R: The reviewer also compares our results to previously published data of Mendell JT et al, 2004 in Nature Genetics. Mendell found 197 and 176 transcripts which are respectively up- and downregulated when NMD is inhibited with an siRNA against UPF1. In this study, the authors used microarray technology and analyzed a pool of ~4,000 transcripts (this amounts to ~9 % deregulation). In our study, we used NGS and analyzed the whole genome. NGS analysis is more sensitive than microarray analysis, which explains why we got a similar number of deregulated genes but with a lower biological impact. This is why $|\log_2FC| \geq 2$ is commonly used, rather than a two-fold change in gene expression.

We have taken into account the observation of reviewer 2 concerning the absence of a DEG list. We have added Supplementary Table 1, with the statistics of the 775 DEGs. We have also added a GO term analysis, provided in Supplementary Table 2, and a schematic representation in Supplemental Figure S2C. We have changed the text accordingly:

First, in HeLa cells, we used RNA-seq to assess the impact of DAP on the whole transcription profile. On the basis of statistics alone (adjusted p-value <0.05), 3,228 differentially

expressed genes (DEGs) were detected, but no major effect of DAP was observed (Fig. S5).

Among the DEGs, 775 showed at least a 2-fold change (adjusted p-value <0.05, $|\log_2FC| \geq 1$

- Table 3)

Reviewer: The authors included the list of genes that were changed by DAP. They have also reduced the threshold.

However, the conclusion of the authors: " Altogether, these results indicate unambiguously that DAP does not interfere with the expression program of HeLa cells at either the mRNA or protein level." is overlooking the effect of DAP on other genes, as expression changes of hundreds of genes is not marginal and raises a significant concern.

R: We modified the text as following: "Altogether, these results indicate that DAP has a moderate impact on the gene expression program and affects mainly pathways to produce energy that could reflect the need for the activation of readthrough."

Reviewer: Changes in hundreds of genes is not "moderate". Thus the conclusion could be for example: "Altogether, these results indicate that DAP has an impact on the gene expression program, affecting mainly pathways to produce energy."

R: We also now include the results of two other toxicity assays: a propidium iodide incorporation assay and the MTT assay. Both assays confirm the absence of any significant toxicity of DAP under our experimental conditions (supplemental figure S4).

Reviewer: The authors now provided the required information.

4. The in vivo experiments in mice presented in Figure 5B are not convincing. The control cells are inadequate, since their tumor development is very different from that in Calu 6 cells with nonsense p53 mutation, as can be seen in the DMSO results. The authors should have chosen cells with the

same tumor growth rate. The concern is that DAP may have an anti-proliferative effect and not a specific effect for UGA nonsense mutations.

R: In order to exclude an antiproliferative effect of DAP, we have repeated the experiment with another cell line (4T1 cells), promoting fast tumor growth (Figure 5A) similar to that promoted by Calu-6 cells. We thank the reviewer for pointing out this possibility.

Reviewer: The addition of 4T1 cells is a good choice.

However, the Western results presented in the original Figure 5C was replaced with another Western, in which the results in lane 7 of DAP, now show a very low effect. What is the significant of these results?

R: I am not sure to understand the comment of the reviewer since on figure 5B, the band intensity in the lane 7 for DAP treatment is quite similar to the band intensity in the lane 6 for instance.

Reviewer: In the original Western blot, in lane 7 (one of the mice) the level of p53 was very high, similar to the level in the control. While in all other lanes (mice) the effect of DAP was marginal.

Since the experiment is new made from new mice samples, we cannot compare the intensity bands with two different experiments. In addition, I would like to remind the reviewer that mice were exposed during a short period of time in the new experiment. Therefore, the rescue of p53 level might be lower than the rescue level observed after several weeks of exposure.

Reviewer: In the new Western, the level of p53 following DAP treatment is very low in all lanes (mice).

Other comments:

1. Page 17 4 lines from the bottom: what does this mean "no visible differences"? is this an adequate way of analyzing the result?

R: We agree with the comment of the reviewer and have modified the sentence as follows: " The two

proteome profiles were found to overlap, without showing any spot shift. This suggests that DAP does not promote synthesis of aberrant proteins (Fig. S6)".

Reviewer: The overlap analysis is insufficiently sensitive to conclude that there are no differences.

R: We now provide a Pearson coefficient which is 0.94. The correlation value ranges between 1 and -1. A value of 1 represents perfect correlation, 0 no correlation, and -1 perfect inverse correlation. In addition, this experiment is a 2D-DIGE experiment which is more sensitive than a 2D gel followed by a coomassie staining provided in other studies (Welch et al., 2007; DOI nature05756 [pii]10.1038/nature05756).

Reviewer: OK

2. Page 18 – the title is misleading since the assay is ex vivo and not in vivo. The ref no 49 is not relevant.

R: This section focuses on mouse experiments, which are in vivo experimentation. The reviewer likely refers to the stability assay on the hepatic extract, which is ex vivo. Since this part of the figure has been moved to "Supporting Information", the title of the section should no longer be confusing. We have also removed the reference 49 and thank the reviewer for pointing out that inaccurate reference.

Reviewer: OK

3. CFTR functional assay – CFTR function in nasal epithelial cells from patients should have been the Ussing Chamber using differentiated cells. This is the standard functional assay to test the effect of drugs on CFTR function. The assay used by the authors, is the SPQ assay. Where the cells differentiated? No details on the analyses is given and reference is to a review paper.

R: For the SPQ assay, we used cells collected from the nose epithelium of patients. We did not

culture them before testing our molecules, in order to test the molecules on the cell model closest to patient cells. Hence, there is no differentiation step before performing the experiment, since the cells are already differentiated. Concerning the reference we used (Mansoura et al., 1999), it is a technical review describing methods for assessing the function of CFTR, including the SPQ assay. We also used this method and described it in the other reference (Benhabiles et al., 2017).

Reviewer: The authors did not perform the required experiment. As explained in the previous comment the authors should have used the gold standard assay, the Ussing chamber measurements for studying the effect of DAP on nonsense mutations in the CFTR gene. This is important in order to compare the effect of DAP to the effect of other tested molecules.

R: To solve this issue that can not be experimentally solved since we do not have anymore these rare patient cells, we removed this experiment.

Reviewer: As said above, it is unfortunate that the CFTR experiments could not be presented with the adequate controls.

4. Figure 2 – the first 3 lanes in each of the panels are unclear. What is the identity of the experiment performed in these three lanes?

R: We have added this information in the figure legend: " The three leftmost lanes show twofold serial dilutions of untreated HeLa cell extract."

Reviewer: The authors now explain what is presented in lanes 1-3. Hence, the results from the Western indicate that DAP has a significant effect when 100-600 μM are used, comparable to the lowest level in HeLa cells. Thus, the correction in Figure 3 using 6.25 or 25 μM , have a low effect on the protein level. This Western should be included in Figure 3, and the results should be discussed accordingly.

R: We now have merged the results of the figure 2 and 3 in order to generate the new figure 2. In addition,

Reviewer: OK

we added in the discussion the following sentences to indicate why we used 25µM of DAP as treatment and why the functional rescue might be suboptimal: " In addition, 25µM was used as the highest concentration of DAP in most of our experiments to characterize the readthrough activity of DAP in order to reduce the risk of an eventual DAP toxicity. As a consequence, the rescue of the function of genes carrying a UGA nonsense mutation by DAP shown in our study (Fig. 2) is likely suboptimal. "

Reviewer: The authors should discuss this point in light of the aim to use DAP in the clinic, since it appears that DAP has a window of concentration that is not toxic and have some effect.

5. The results presented in Figure 3B and C are inconsistent with the results presented in Figure 2. This includes the G418 different concentrations as well as DAP in comparison to G418. For example, in Figure 3B and C, there is no difference in the effect of G418 using 25 or 600 µM whereas in the Western blot presented in Figure 2, there is a significant difference in p53 expression between these concentrations. More importantly, 25µM DAP has the same effect on the p53 protein level as 600µM G418 whereas in Figure 3B and C the effect of these two treatments is very different. These results are confusing. Can the authors explain them?

R: The results in Figure 2 show that both DAP and G418 are able to promote the synthesis of fulllength p53 protein, but these data provide no information about the functionality of the fulllength p53 protein synthesized. Figures 3B and 3C aim to demonstrate whether the full-length p53 protein is functional or not. Our results show that DAP promotes synthesis of a functional full-length p53 protein; G418, in contrast, promotes synthesis of a non-functional full-length p53 protein.

Reviewer: The authors do not explain the inconsistent results detailed in this comment. Moreover, it

is puzzling that G418 readthrough results in a non-functional proteins whereas DAP results in a functional protein, as the two reagents presents the same protein level

R: I apologize for not being clear. My point is that the synthesis of a full length protein is not a warranty of producing a functional protein because the amino-acid introduce at the PTC position might not be compatible with the function of the protein. G418 and DAP have different mode of action and when DAP will promote the insertion of a tryptophan, G418 might induce the introduction of another amino acid at that position making the full-size protein a not functional protein. It is not a question of amount of protein but more about the composition in amino acid that will make the difference in figure 2 panels C, D and E.

Reviewer: The authors should have performed experiments to investigate the amino acids introduced into the protein by G418 readthrough, as they performed in figure 5A for DAP.

6. Figure 5C – at what time point was the analyses performed?

What is the 1-7? Different mice? Different days?

R: The tumor size measurements were performed three times per week and the western-blot analysis was performed on tumors collected at the time of sacrifice. We have modified the figure to clarify this and changed the X-axis unit into days after cell injection.

Reviewer: OK

7. Figure 5C shows a very low level of WT p53 in the DAP experiment. This may indicate that the tumor development results are due to a combined effect of DAP on UGA stop and its activity as an anti-proliferative agent.

R: We do not favor the hypothesis of an anti-proliferative effect of DAP on tumor growth, since we observe no such anti-proliferative effect on cells carrying no nonsense mutation (i.e. the HCC-GR6

cells in the previous version of the manuscript and the 4T1 cells in the new version of the manuscript). The low level of p53 detected in tumors is also due to the fact that the TP53 gene is highly regulated at different levels of the gene expression pathway. Although we were able to detect this low-level expression of p53 in mice exposed to DAP, we were unable to detect any expression in mice exposed to DMSO, either after a long exposure period (Figure 5C in the initial version of the manuscript) or after a short period of exposure beginning after the tumor had already developed (new figure 5B)).

Reviewer: The revised figure 5 is lacking a control. The authors should include as a control 4T1.

R: the reason why 4T1 cells were not included in the p53 western-blot is that the antibody used to detect p53 recognizes only human p53 and not mouse p53 in order to prevent the detection of any mouse tissue contamination in the tumor analysis as requested. In addition, 4T1 cells do not express p53 protein.

Reviewer: The authors did not perform the required experiments. Thus, the magnitude of effect of DAP on p53 cannot be evaluated.

8. Figure 5 – the numbers in X axis should be explained. Are these time points of analysis? There were 3 analysis a week for 21 days (according to the text). What are the 17 data points?

R: The reviewer is right, our explanation was not clear and the experiment was not performed for 21 days but for about 5 weeks. We apologize for this and have corrected the text.

Reviewer: OK

9. Proteome analysis – the authors should perform the proteomic analysis on Calu6 and not on HeLa. In order to be able to identify the change in p53.

R: We purposely performed the experiment on cells that do not have nonsense mutations in order to

exclude any change in proteomic profile due to re-expression of a mutant gene. By doing the experiment on Calu-6, we would induce expression of the TP53 gene and affect the expression of many other genes. We thus think that the best cell line to use is HeLa and not Calu-6.

Reviewer: As all the experiments were performed in Calu-6, the proteome analysis should have been performed in these cells.

R: This experiment was performed in order to determine whether DAP interfere with the general protein expression profile. Making the proteome analysis in Calu-6 would make the analysis and the conclusion inaccurate because of the rescue of TP53 expression would impact the expression of many genes. We would expect to have a modification of the proteome profile due to this p53 expression rescue and not due to the general readthrough activity of DAP which is the aim of this experiment. We therefore chose to perform it in HeLa cells as we did to determine the impact of DAP on the general transcription profile (Fig. S5).

Reviewer: Again, the authors did not perform the required experiments. There are experimental and bioinformatics solutions to the p53 effect on gene expression. For example, to exclude changes due to p53 activation, the cells could be exposed to other treatment inducing p53.

10. The manuscript would benefit from English editing

R: the manuscript was originally corrected for English writing and has received a second round of correction before submission of the revised version of the manuscript.

Reviewer: OK

One more time, we would like first to thank the reviewer for its comments and time.

Reviewer #2 (Remarks to the Author):

We would like to thank the three reviewers for their time and their comments on the revised version of our manuscript. Here are our answers that we hope will address their last concerns.

Reviewers' comments:

Reviewer 2

1. The functional assays are lacking WT controls as a reference. Without comparison to the WT function, the efficiency and level of correction cannot be concluded. This should be performed for the p53 and the CFTR analyses. Only in the mice experiments (Fig 5) a WT p53 control level is shown.

R: We have now included a p53 functional assay with HeLa cells (new figures 3B and 3C). Results are presented in separate panels, as Calu-6 and HeLa cells are not directly comparable because of different genetic backgrounds. For the CFTR assay, our authorization was limited to cystic fibrosis patients carrying nonsense mutations on both alleles encoding CFTR, and did not include the authorization to collect cells from healthy persons.

Reviewer: In order to evaluate the effect magnitude of DAP on readthrough of nonsense mutations, a comparison to WT activity is required.

For p53, the authors now added results from HeLa cells, that do not have a p53 nonsense mutation. The results showed no additional p53 activity following DAP treatment, as expected. However, still there is no way to evaluate the magnitude of DAP effect in Calu-6 cells.

Unfortunately, also the authors did not perform the required additional experiments for CFTR. Thus, evaluation of the effect in Figure 3D is not possible. Furthermore, the results are confusing since patient 1 and 2 differ in their response despite carrying the same genotype (that surprisingly, was omitted from the revised Figure 3D).

R: We totally understand the issue raised by the reviewer. We removed the results concerning the rescue of the function of CFTR in patient cells since our authorization ended and did not cover the collect of cells from healthy people.

Reviewer: It is unfortunate that no control experiments could be performed in CFTR mutated cells. Having said that, without appropriate controls the results can not be evaluated.

Concerning the cancer cell lines we added the following sentence to the discussion since the concern raised by the reviewer is impossible to address due to the lack of isogenic wild-type cells and the use of transfected construct would be inaccurate due to variability of gene expression:

“Although DAP appears to be more potent than G418, we were not able to calculate the rescue of the gene expression compared to wild-type since no isogenic wild-type cells corresponding to our used cell lines exist.”

Reviewer: Thus, the effect of DAP in cells carrying a p53 nonsense mutation cannot be evaluated.

R: We never evaluated the rescue of the mutant gene expression to the expression of the wild-type version of the gene since it is known that it is variable from one person to another or from one cell line to another one. As explained before, cell models fitting these criteria do not exist to our knowledge. We always compared the efficacy of DAP to the level of the gene expression in the presence of DMSO as a negative control and to the efficacy of nonsense mutation correction in the presence of G418 which was known as the most potent readthrough molecule. Our conclusions always been to say that DAP has a more potent readthrough efficacy on UGA nonsense mutations than G418. Our results show clearly the rescue of the p53 expression in the presence of DAP since

we do not detect p53 protein when cells are treated with DMSO and the p53 protein is detected after DAP treatment for instance. In addition, the amount of p53 protein obtained upon DAP treatment are higher than the amount of p53 protein obtained upon G418 treatment.

2. As was shown in the past, each cell type/line has a different NMD efficiency, which affects the level of read-through. The effect of NMD efficiency in each of the studied cellular systems is missing. The authors should include experiments with either NMD inhibitors or siRNA against key NMD factors.

R: The reviewer is right about the influence of NMD on readthrough. We show that DAP does not inhibit NMD (figure 4A). We were careful not to compare the readthrough efficacy of DAP between different cell types. We agree that to do that we would need to normalize the NMD efficiency, probably by inhibiting it, but this will be the focus of another study showing the benefit of using a combination of NMD inhibitors and readthrough activators.

Reviewer: Unfortunately, the authors did not perform the required experiments.

R: The experiments suggested by the reviewer are very interesting but are out of the scope of this study that focuses on the characterization of DAP as a readthrough activator. Our study does not focus on the optimization of the efficiency of DAP that is already very high. The reviewer experiments would require to identify an NMD inhibitor compatible with DAP treatment and we know that only few NMD inhibitors have been identified since we are actively involved in this domain too. Then, the combination DAP with NMD inhibitor would have to be used to test on cell models and in vivo in mouse model in order to get the information about whether or not it is possible to obtain a higher rescue efficiency on the expression of PTC-containing genes. This information is not necessary for this first study on DAP that has to show its capacity at correcting UGA nonsense mutation by itself. The combination will be included in a future study focusing on the therapeutic aspects of DAP and the way to optimize the correction of nonsense mutation by DAP.

Reviewer: It is not clear why the authors claims that NMD inhibitor should be compatible with DAP. Several NMD inhibitors are available and widely used among them: SMG1 inhibitor, NMD1, siRNA against key NMD factors among others. The NMD experiments are expected to significantly add to the manuscript by revealing the total effect of DAP.

R: The focus of our manuscript is to demonstrate that DAP has readthrough capacity on UGA nonsense mutations and to explain how the molecule works. The evaluation of the best way to correct a UGA nonsense mutation by using DAP and an NMD inhibitor will be the focus of another study. This last point is too important to be treated in one figure and would require a complete study.

4. The in vivo experiments in mice presented in Figure 5B are not convincing. The control cells are inadequate, since their tumor development is very different from that in Calu 6 cells with nonsense p53 mutation, as can be seen in the DMSO results. The authors should have chosen cells with the

same tumor growth rate. The concern is that DAP may have an anti-proliferative effect and not a specific effect for UGA nonsense mutations.

R: In order to exclude an antiproliferative effect of DAP, we have repeated the experiment with another cell line (4T1 cells), promoting fast tumor growth (Figure 5A) similar to that promoted by Calu-6 cells. We thank the reviewer for pointing out this possibility.

Reviewer: The addition of 4T1 cells is a good choice.

However, the Western results presented in the original Figure 5C was replaced with another Western, in which the results in lane 7 of DAP, now show a very low effect. What is the significant of these results?

R: I am not sure to understand the comment of the reviewer since on figure 5B, the band intensity in the lane 7 for DAP treatment is quite similar to the band intensity in the lane 6 for instance.

Reviewer: In the original Western blot, in lane 7 (one of the mice) the level of p53 was very high, similar to the level in the control. While in all other lanes (mice) the effect of DAP was marginal.

Since the experiment is new made from new mice samples, we cannot compare the intensity bands with two different experiments. In addition, I would like to remind the reviewer that mice were exposed during a short period of time in the new experiment. Therefore, the rescue of p53 level might be lower than the rescue level observed after several weeks of exposure.

Reviewer: In the new Western, the level of p53 following DAP treatment is very low in all lanes (mice).

R: The results of the two western-blot that the reviewer wants to quantitatively compare are from two in vivo experiments made at about one year difference. From my point of view, it is just possible to conclude that DAP is able to rescue the expression of p53 gene at the protein level in vivo in both experiments. The level of a gene expression in vivo can be influenced by many parameters and in particular from one mouse to another one. I agree that it seems the rescue might be less efficient but this rescue occurs in all treated mice as it was for the previous experiment. Both experiments support our conclusions.

3. CFTR functional assay – CFTR function in nasal epithelial cells from patients should have been the Ussing Chamber using differentiated cells. This is the standard functional assay to test the effect of drugs on CFTR function. The assay used by the authors, is the SPQ assay. Where the cells differentiated? No details on the analyses is given and reference is to a review paper.

R: For the SPQ assay, we used cells collected from the nose epithelium of patients. We did not culture them before testing our molecules, in order to test the molecules on the cell model closest to patient cells. Hence, there is no differentiation step before performing the experiment, since the cells are already differentiated. Concerning the reference we used (Mansoura et al., 1999), it is a technical review describing methods for assessing the function of CFTR, including the SPQ assay. We also used this method and described it in the other reference (Benhabiles et al., 2017).

Reviewer: The authors did not perform the required experiment. As explained in the previous comment the authors should have used the gold standard assay, the Ussing chamber measurements for studying the effect of DAP on nonsense mutations in the CFTR gene. This is important in order to compare the effect of DAP to the effect of other tested molecules.

R: To solve this issue that can not be experimentally solved since we do not have anymore these rare patient cells, we removed this experiment.

Reviewer: As said above, it is unfortunate that the CFTR experiments could not be presented with the adequate controls.

R: When we designed the experiment, we wanted to demonstrate that DAP can rescue the function of CFTR in comparison to the same patient cells treated with DMSO. We succeed to show it. We did not include a healthy control because we estimated that would be wrong to compare with another person for which the level of CFTR gene expression would be different from the CFTR gene expression of cystic fibrosis patient cells even at the transcription level.

5. The results presented in Figure 3B and C are inconsistent with the results presented in Figure 2. This includes the G418 different concentrations as well as DAP in comparison to G418. For example, in Figure 3B and C, there is no difference in the effect of G418 using 25 or 600 μ M whereas in the Western blot presented in Figure 2, there is a significant difference in p53 expression between these concentrations. More importantly, 25 μ M DAP has the same effect on the p53 protein level as 600 μ M G418 whereas in Figure 3B and C the effect of these two treatments is very different. These results are confusing. Can the authors explain them?

R: The results in Figure 2 show that both DAP and G418 are able to promote the synthesis of fulllength p53 protein, but these data provide no information about the functionality of the fulllength p53 protein synthesized. Figures 3B and 3C aim to demonstrate whether the full-length p53 protein is functional or not. Our results show that DAP promotes synthesis of a functional full-length p53 protein; G418, in contrast, promotes synthesis of a non-functional full-length p53 protein.

Reviewer: The authors do not explain the inconsistent results detailed in this comment. Moreover, it is puzzling that G418 readthrough results in a non-functional proteins whereas DAP results in a functional protein, as the two reagents presents the same protein level

R: I apologize for not being clear. My point is that the synthesis of a full length protein is not a warranty of producing a functional protein because the amino-acid introduce at the PTC position might not be compatible with the function of the protein. G418 and DAP have different mode of action and when DAP will promote the insertion of a tryptophan, G418 might induce the introduction of another amino acid at that position making the full-size protein a not functional protein. It is not a question of amount of protein but more about the composition in amino acid that will make the difference in figure 2 panels C, D and E.

Reviewer: The authors should have performed experiments to investigate the amino acids introduced into the protein by G418 readthrough, as they performed in figure 5A for DAP.

R: The identification of the amino-acid incorporated in the presence of DAP at the UGA position supports the idea that DAP interferes with the activity of the FTSJ1 enzyme. It was one of the experiments to investigate the mode of action of DAP. The comparison with G418 activity would not bring any information on the mode of action of DAP explaining why we did not investigate the amino-acid incorporated in the presence of G418.

7. Figure 5C shows a very low level of WT p53 in the DAP experiment. This may indicate that the tumor development results are due to a combined effect of DAP on UGA stop and its activity as an anti-proliferative agent.

R: We do not favor the hypothesis of an anti-proliferative effect of DAP on tumor growth, since we observe no such anti-proliferative effect on cells carrying no nonsense mutation (i.e. the HCC-GR6 cells in the previous version of the manuscript and the 4T1 cells in the new version of the manuscript). The low level of p53 detected in tumors is also due to the fact that the TP53 gene is highly regulated at different levels of the gene expression pathway. Although we were able to detect this low-level expression of p53 in mice exposed to DAP, we were unable to detect any expression in mice exposed to DMSO, either after a long exposure period (Figure 5C in the initial version of the manuscript) or after a short period of exposure beginning after the tumor had already developed (new figure 5B)).

Reviewer: The revised figure 5 is lacking a control. The authors should include as a control 4T1.

R: the reason why 4T1 cells were not included in the p53 western-blot is that the antibody used to detect p53 recognizes only human p53 and not mouse p53 in order to prevent the detection of any mouse tissue contamination in the tumor analysis as requested. In addition, 4T1 cells do not express p53 protein.

Reviewer: The authors did not perform the required experiments. Thus, the magnitude of effect of DAP on p53 cannot be evaluated.

R: We agree with the reviewer that we cannot evaluate the magnitude of the effect of DAP on p53 rescue. We never indicate this magnitude for the reasons exposed before.

One more time, we would like to thank this reviewer for the interesting points that he/she raised on our study.